# Upregulated energy metabolism in the Drosophila mushroom body is the trigger for long-term memory

Pierre-Yves Plaçais[1], Éloïse de Tredern[1,*], Lisa Scheunemann[1,*], Séverine Trannoy[1,†], Valérie Goguel[1], Kyung-An Han[2], Guillaume Isabel[1,†] & Thomas Preat[1]

Efficient energy use has constrained the evolution of nervous systems. However, it is unresolved whether energy metabolism may resultantly regulate major brain functions. Our observation that Drosophila flies double their sucrose intake at an early stage of long-term memory formation initiated the investigation of how energy metabolism intervenes in this process. Cellular-resolution imaging of energy metabolism reveals a concurrent elevation of energy consumption in neurons of the mushroom body, the fly's major memory centre. Strikingly, upregulation of mushroom body energy flux is both necessary and sufficient to drive long-term memory formation. This effect is triggered by a specific pair of dopaminergic neurons afferent to the mushroom bodies, via the D5-like DAMB dopamine receptor. Hence, dopamine signalling mediates an energy switch in the mushroom body that controls long-term memory encoding. Our data thus point to an instructional role for energy flux in the execution of demanding higher brain functions.

[1] Genes and Dynamics of Memory Systems, Brain Plasticity Unit, CNRS, ESPCI Paris, PSL Research University, 10 rue Vauquelin, Paris 75005, France. [2] Department of Biological Sciences, Border Biomedical Research Center, University of Texas at El Paso, El Paso, Texas 79968, USA. † Present addresses: Department of Neurobiology, Harvard Medical School, Boston, Massachusetts 02115, USA (S.T.); Research Center on Animal Cognition, Université Paul Sabatier, Toulouse, France (G.I.). * These authors contributed equally to this work. Correspondence and requests for materials should be addressed to P.-Y.P. (email: pierre-yves.placais@espci.fr) or to T.P. (email: thomas.preat@espci.fr).

Energy fluxes, that is, the sequential catabolism of energy-carrying molecules, involve substrates that are extremely conserved across the animal kingdom, even more so than genes or signalling pathways. This is especially striking for central nervous systems, which use glucose as their main energy source[1]. Energy efficiency is indeed put forward as a major factor of the selective pressure driving the evolution of nervous systems[2]. Moreover, efficient power use is stated as a design principle of neuronal network architecture[3]; this underlies for example the widespread occurrence of sparse coding in sensory systems[3–5]. Beyond network architecture, one pertinent question is how energy fluxes intervene in brain function. The predominant view in the field of neuroenergetics is that energy is supplied 'on demand' to neurons in support of their activity[6]. However, some data also suggest active regulation of brain function, especially memory, by glucose[7]. Yet, our knowledge about the interplay between energy metabolism and memory formation at the molecular and cellular levels remains very limited. Several studies documented the importance of astrocytic lactate production for memory[8,9], especially long-term memory[10], but there has been much evidence as well that lactate may have signalling roles aside from being an energy substrate[11–15]. Hence, the major question remains open whether the magnitude of energy flux could be informative for neurons, wherein it would control and not only support memory processes or other brain functions.

Drosophila is a genetically tractable organism, and as such it is a powerful animal model to address these questions. Flies can form olfactory memory as a result of the association between an odorant and electric shocks. Experiencing a single cycle of olfactory training is not sufficient for the flies to form long-lasting memory (Fig. 1a). A repeated massed training protocol yields a 24 h-memory called anaesthesia-resistant memory (ARM), which does not rely on de novo protein synthesis[16]. The most stable long-term memory (LTM), which is protein synthesis-dependent, requires rest intervals between the multiple associations (spaced training)[16]. Therefore, the LTM formation does not simply result from accumulating knowledge through repeated trials, but rather relies on specific mechanisms that are triggered upon the particular temporal pattern of spaced training[17]. We previously showed that the trigger for LTM formation involves rhythmic signalling on mushroom body (MB) neurons from specific dopaminergic neurons, which occurs during and immediately after spaced training[18]. The MBs are the major integrative brain center that supports learning and memory in insects, and a putative functional homologue of the mammalian hippocampus[19,20]. Drosophila MBs are paired structures including ~2,000 intrinsic neurons per brain hemisphere. These neurons receive dendritic input from the antennal lobes through projection neurons[21] in the calyx area on the posterior part of the brain. Their axons form a fascicle that traverses the brain to the anterior part, where they branch to form horizontal and vertical lobes according to three major branching patterns[22]. MB lobes are covered by compartmentalized dopaminergic innervation from 20 cell types, subsets of which provide reinforcement signalling during olfactory conditioning[23–25], through the dopR/dDA1 receptor[26,27]. MB neurons can therefore detect the coincident onset of olfactory and electric shock stimuli[28,29].

We previously showed that LTM formation is blocked when flies are starved before and after training, which is beneficial to their survival to food restriction[30]. This prompted us to investigate the interplay between LTM formation and energy metabolism in Drosophila, through a combination of behavioral experiments involving genetic knock-down or neural circuit manipulation in adult flies, in vivo imaging of pyruvate metabolism at the cellular level, and feeding assays. Our results establish that following spaced training, flies strongly increase their energy intake. This reflects an elevation of energy flux in MB neurons, which we show is a necessary, but also a sufficient condition to consolidate memory into LTM. This upregulation is initiated by dopamine signalling through a specific receptor, the D5-like DAMB receptor. Dopamine signalling thus mediates an energy-based gating that is instructive to initiate LTM formation in the Drosophila MB.

## Results

**Flies eat more sucrose upon LTM formation.** First, we sought to evaluate whether LTM formation is a costly process, and if so, to what extent. Flies were subjected to a spaced training protocol, and their subsequent intake of a sucrose solution was measured using two distinct methods: a dye-feeding assay and a Capillary Feeding (CAFE) assay (see Experimental Procedures for details). Immediately after spaced training, flies consumed twice as much sucrose solution as control flies that had undergone a non-associative unpaired protocol (Fig. 1b,c). This effect did not occur following the massed training protocol (Fig. 1b,c). The CAFE assay additionally demonstrated that the increased feeding behaviour specifically was observed only during the 4 h following spaced training (Fig. 1c). Flies had direct and ad libitum access to pure mineral water in this assay, suggesting that their increased feeding behaviour after spaced training results from a search to increase their energy intake, and not from elevated thirst. To confirm this, we conducted a control dye-feeding assay without sucrose. In this case, there was no difference in dye ingestion between flies that underwent a spaced training or an unpaired protocol (Fig. 1d). Finally, we asked if the increased feeding could be the result of higher locomotor activity. We monitored the locomotor activity of flies for 4 h after either of these two protocols (see Experimental Procedures for details). During the first hour, and also during the whole period of measurement, we observed no difference between the two conditions (Fig. 1e), which indicates that the increased energy intake after spaced training does not stem from an elevated energy demand due to increased activity, but is actually due to LTM formation itself. This series of experiments, performed on flies with unlimited access to food before conditioning, strongly suggests that spaced training has a dramatic impact on the energy budget of the whole fly. Surprisingly, regulation of energy metabolism starts at an early stage in LTM formation.

**MB energy metabolism is increased upon LTM formation.** To confirm the impact of LTM formation on energy balance, we aimed to detect whether an upregulation in energy consumption actually occurred in the fly brain following spaced training, and more specifically in MB neurons, which play a prominent role in olfactory LTM[31–33]. In vivo imaging of energy metabolism with cellular resolution has become possible only recently, thanks to the development of genetically-encoded fluorescent indicators that are sensitive to the intracellular concentrations of various energy metabolites[34]. Both glucose and lactate can act as neuronal energy sources, but in either case the energy-producing pathways converge to yield pyruvate, the input energy substrate for oxidative phosphorylation in mitochondria. We introduced the pyruvate sensor Pyronic[35] into Drosophila, which was expressed in MB neurons using the UAS/GAL4 system, by means of the VT30559 GAL4 driver, which is very specific to MB neurons within the central brain (Supplementary Fig. 1A, Supplementary Movies 1 and 2). The Pyronic ratio (see Experimental Procedures), which increases with pyruvate concentration, was recorded by two-photon imaging from the MB vertical lobes, an established site of LTM encoding[31,32] (Fig. 2a). Although FRET

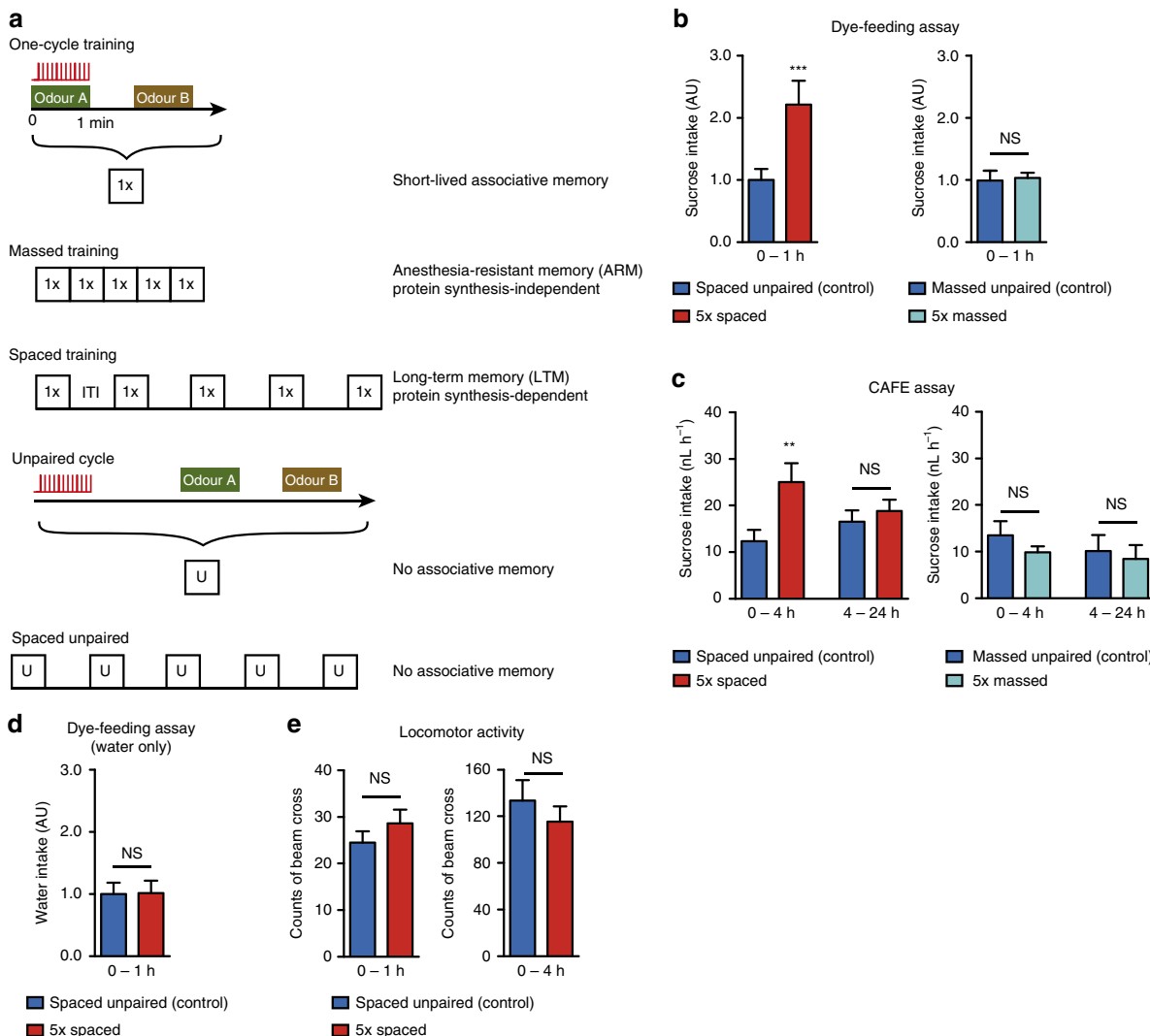

**Figure 1 | Spaced training causes increased sucrose intake.** (**a**) Diagram illustrating the different olfactory aversive conditioning protocols used in this study (ITI: inter-trial intervals, 15-min duration). For the so-called massed unpaired control (**b,c**), a reduced ITI of 2 min was applied to avoid association between shocks and the odour B of the previous cycle. (**b**) Flies were transferred onto a dyed 5% sucrose solution for 1 h immediately after the indicated conditioning procedure. Following a spaced associative training, flies ingested twice as much sucrose as the non-associative spaced unpaired control ($n = 17$ groups of five flies; $t_{32} = 4.18$; $P = 0.0002$). This was not the case following a massed training ($n = 12$; $t_{22} = 0.19$; $P = 0.85$). (**c**) Sucrose intake by single flies was measured 4 and 24 h after the indicated conditioning procedure in a CAFE assay. During the first 4 h (but not after) flies ingested twice as much sucrose after their spaced training as after the unpaired control protocol (0–4 h: $n = 111$–112; $t_{221} = 2.69$; $P = 0.007$. 4–24 h: $n = 110$–112; $t_{220} = 0.65$; $P = 0.51$). No difference was observed at any time point following massed protocols (0–4 h: $n = 32$; $t_{62} = 1.11$; $P = 0.27$, 4–24 h: $n = 31$–32; $t_{61} = 0.37$; $P = 0.71$). (**d**) In a dye-feeding assay without sucrose, there were no difference between flies of the spaced and unpaired conditions ($n = 16$ groups of 5 flies; $t_{30} = 0.058$; $P = 0.95$). (**e**) The locomotor activity of flies was measured as the number of beam crosses in a Trikinetics device during 4 h following conditioning. Results are displayed for the first hour after conditioning, as a control for the dye-feeding assay or for the whole period, as a control for the CAFE assay. In both cases, there was no difference between flies that had a spaced training or an unpaired control protocol ($n = 130$–131; 0–1 h: $t_{253} = 1.08$; $P = 0.28$; 0–4 h: $t_{253} = 0.87$; $P = 0.39$).

sensors are sensitive to the intracellular concentration, what is physiologically significant is the energy flux. Hence the relevant measurement is the rate of pyruvate consumption or production, rather than its concentration. Inspired by a previously published protocol[35], we used sodium azide (a potent inhibitor of mitochondrial complex IV) to block pyruvate mitochondrial uptake and measure the resulting pyruvate accumulation. In naive flies, the addition of 5 mM azide to the solution bathing the brain during live imaging induced a strong and rapid increase in the Pyronic ratio (Fig. 2b, Supplementary Fig. 1B), until saturation of the sensor. We performed similar experiments in flies that had been subjected to different conditioning protocols, in a time

window (30 min to 2.5 h after the end of conditioning) when flies display increased sucrose intake after spaced training (see Fig. 1c). Compared to an unpaired spaced protocol or to a massed training, the spaced training resulted in a marked enhancement of pyruvate consumption in MB neurons (Fig. 2c,d, Supplementary Fig. 1C). After unpaired or massed protocols, pyruvate consumption rates were similar to those measured in naive flies.

On cultured mammalian astrocytes, it was shown that treatment with rotenone, another inhibitor of oxidative phosphorylation, causes an activation of glycolysis in reaction to mitochondrial blocking[36]. Thus, the pyruvate accumulation following azide

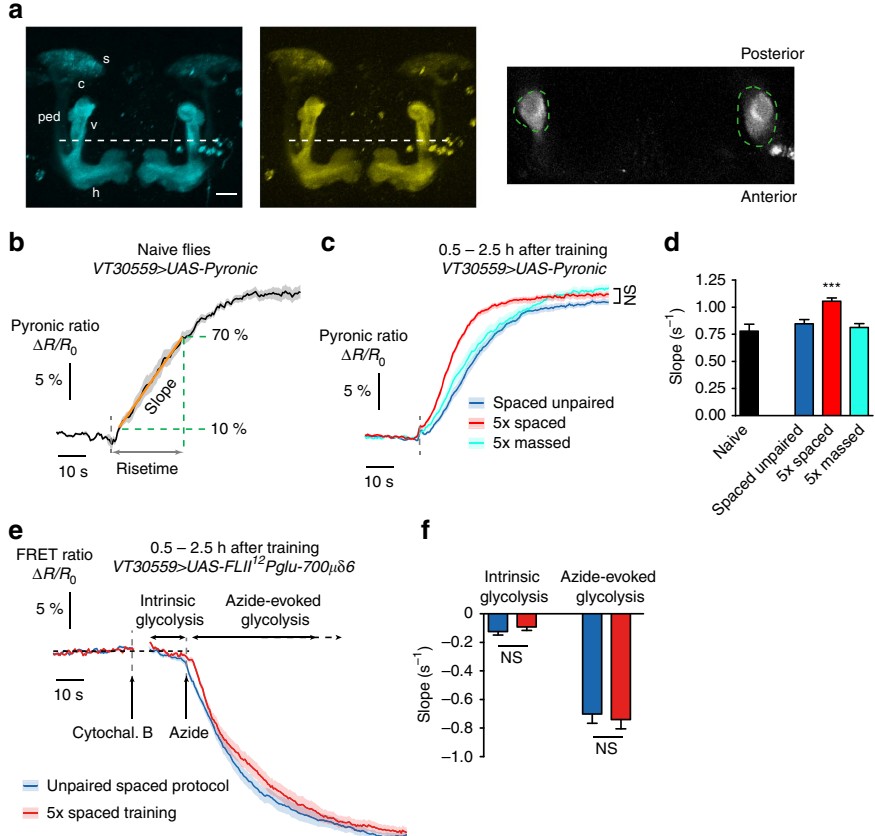

**Figure 2 | The energy flux in MB neurons is increased following spaced training.** (**a**) Images of the Pyronic sensor expressed in MB neurons through the *VT30559* driver, obtained by 2-photon microscopy. The two images on the left show the 3D reconstruction of image stacks in the mTFP and Venus channels in the conditions that were used for live recordings (scale bar: 20 μm). MB neurons have their soma (s) in the posterior part of the brain; their dendrites form the calyx (c), and axons are bundled into the peduncle (ped) that traverses the brain posteriorly to anteriorly, where axons ramify into vertical (v) and horizontal (h) lobes. Live recordings presented thereafter were obtained from horizontal planes located halfway to the top of the vertical lobes (dashed white line). An illustrative example is shown on the right, with the corresponding regions of interest (green dashed lines). (**b**) Application of 5 mM sodium azide (grey dashed line) results in an increased Pyronic ratio in the MB vertical lobes of naive flies ($n = 6$). The rise was linear up to 70% of the plateau. The slope and risetime were used to quantify the kinetics of pyruvate accumulation during this linear phase (see Experimental Procedures). (**c**) Measurement of pyruvate accumulation with the same procedure after different conditioning protocols (unpaired: $n = 22$; $5 \times$ spaced $n = 26$; 5x massed $n = 16$). There was no difference in the plateau value between the three conditions ($F_{2,121} = 3.05$, $P = 0.051$). (**d**) Average slope from data shown in **b**,**c**. The slope after spaced training was higher than after a spaced unpaired protocol or after massed training ($F_{2,121} = 15.37$, $P < 0.0001$). (**e**) The FRET glucose sensor FLII[12]Pglu-700μδ6 was expressed in MB neurons to monitor the azide-induced activation of glycolysis in flies that had undergone spaced training or unpaired control protocol ($n = 15$ per condition). Intrinsic glycolysis was measured following the application 20 μM cytochalasin B, a pharmacological blocker of glucose transporter (see Experimental Procedures for details). A subsequent application of 5 mM azide strongly enhanced glucose consumption. (**f**) There was no difference between flies of the two conditions either for intrinsic glycolysis rate ($t_{56} = 0.99$; $P = 0.32$) or for azide-evoked glycolysis ($t_{56} = 0.43$; $P = 0.67$).

treatment that we measured in MB neurons may originate not only from the stop of mitochondrial uptake, but also from the increase of pyruvate production by glycolysis. To clarify the interpretation of our Pyronic data, we expressed a FRET glucose sensor FLII[12]Pglu-700μδ6 in MB neurons to monitor the effect of azide treatment on glycolysis, in a protocol similar to ref. 36. We observed that azide treatment indeed instantly boosted glucose consumption (Fig. 2e). Interestingly, comparing flies after spaced training or after unpaired protocol, we observed that both intrinsic and azide-evoked glycolytic rates were similar between the two conditions (Fig. 2e,f). This indicates that the accelerated azide-evoked pyruvate accumulation observed in flies after spaced training does not come from an upregulated glycolytic capacity, but from an increased pyruvate production from another source, that is not any more consumed by mitochondria. Altogether, this series of experiments points to an upregulation of mitochondrial flux in MB neurons shortly after spaced training.

**Energy metabolism is not globally upregulated.** The upregulation of energy metabolism that we evidenced in MB neurons could be part of a more global shift in brain energy consumption. To address this point, we measured the rate of pyruvate accumulation induced by azide in other brain structures. We expressed the Pyronic sensor through the *Feb170* driver, which was used previously to target the ellipsoid body (EB)[37], and which also labels a cluster of median neurosecretory cells (mNSC) in the pars intercerebralis (Fig. 3a). In these two structures, azide application resulted in an accumulation of pyruvate (Fig. 3b). Noticeably, the plateau value reached by the Pyronic ratio was consistently lower in EB neurons than in mNSC (Fig. 3b) (see Discussion). Interestingly, in both structures, the rate of pyruvate accumulation after spaced training or after an unpaired protocol was similar (Fig. 3c–e). This shows that energy metabolism is not upregulated in the whole brain upon LTM formation, and that it might be an MB-specific phenomenon.

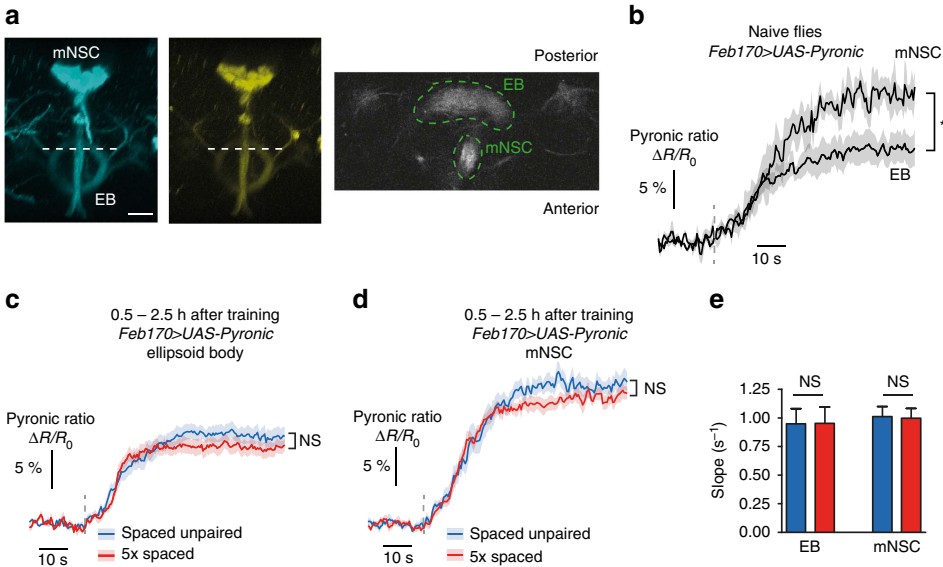

**Figure 3 | The energy flux in ellipsoid body neurons or in median neurosecretory cells is not increased following spaced training.** (**a**) Images of the Pyronic sensor expressed through the *Feb170* driver, obtained by two-photon microscopy. The two images on the left show the 3D reconstruction of image stacks in the mTFP and Venus channels in the conditions that were used for live recordings (scale bar, 20 μm). This driver is commonly used to label ellipsoid body (EB) neurons, but it also strongly labels a cluster of ~20 median neurosecretory cells (mNSC). Live recordings were obtained from a horizontal plane where both the upper part of the EB and a section of the descending branch from mNSC were visible (dashed white line). An illustrative example is shown on the right, with the corresponding regions of interest (green dashed lines). (**b**) Application of 5 mM sodium azide (grey dashed line) caused an increase in the Pyronic ratio in the EB and the mNSC ($n = 5$). The plateau value was significantly lower in the EB than in the mNSC ($t_9 = 2.62$; $P = 0.028$). (**c**) Measurement of pyruvate accumulation in EB neurons following azide application after spaced training or control unpaired protocol. The plateau value was not statistically different between the two conditions ($n = 12$–13; $t_{23} = 1.47$; $P = 0.15$). (**d**) Measurement of pyruvate accumulation in mNSC following azide application after spaced training or control unpaired protocol. The plateau value was not statistically different between the two conditions ($n = 13$; $t_{24} = 1.37$; $P = 0.18$). (**e**) Average slope from data shown in **c**,**d**. The slope after spaced training was not different from after a spaced unpaired protocol in either structure (EB: $t_{23} = 0.06$; $P = 0.96$; mNSC: $t_{24} = 0.12$; $P = 0.90$).

**Increased energy flux in MB is sufficient for LTM formation.** The mitochondrial pyruvate dehydrogenase (PDH) complex catalyses the first step of pyruvate metabolism for oxidative phosphorylation. The activity of the E1 enzyme of this complex (PDHE1) is regulated, being activated by the pyruvate dehydrogenase phosphatase (PDP)[38,39], and inhibited by phosphorylation through pyruvate dehydrogenase kinase (PDK; Fig. 4a)[39]. To investigate whether enhanced energy flux actually underlies LTM formation, we aimed to manipulate the level of mitochondrial activity in MB neurons. We expressed exclusively at the adult stage (see Experimental Procedures) RNAi against the β subunit of PDHE1 (PDHE1β), PDP or PDK in MB neurons together with the Pyronic sensor. Following azide application, the rate of pyruvate accumulation was decreased in naive flies expressing either PDHE1β or PDP RNAi compared to control flies, and was conversely increased in flies expressing PDK RNAi (Fig. 4b). The plateau value of the Pyronic ratio was also lower with PDHE1 and PDP RNAi (Fig. 4b), betraying higher steady-state pyruvate level (Supplementary Fig. 2A), probably because glycolysis was more activated in these flies as a result of the reduction of mitochondrial activity. These results are thus consistent with PDHE1 and PDP knockdown powering down mitochondrial flux, and on the contrary PDK knockdown boosting it.

Interestingly, the expression of RNAi against PDHE1β or PDP in MB neurons at adult stage impaired LTM formation following spaced training (Fig. 4c,d), without altering the naive odour or electric shock avoidance (Supplementary Table 1). No defect was observed in the absence of RNAi induction (Supplementary Fig. 2B,C). Importantly, the memory induced by massed training was unaffected by the expression of either RNAi (Fig. 4c,d).

Hence, spaced training specifically triggers an increase in MB energy consumption, which is pivotal for LTM formation.

We next asked whether the increased energy consumption could act as a trigger for LTM encoding. If so, we reasoned that forced enhancement of MB energy flux should facilitate the consolidation of the learned experience into LTM. On the contrary, it should not facilitate LTM formation if the increased energy flux is only supportive of LTM but does not constitute a signal that triggers the LTM pathway. LTM formation normally requires at least five spaced cycles[16–18]. We trained flies with only two spaced cycles of associative conditioning and measured the memory score 24 h later. Flies that expressed RNAi against PDK in MB neurons at the adult stage exhibited higher memory than their controls (Fig. 4e, Supplementary Fig. 2D), but normal odour or shock avoidance (Supplementary Table 1). Moreover, the memory formed by these flies was sensitive to the ingestion of cycloheximide (CXM), a protein synthesis inhibitor, before conditioning, which is a hallmark of LTM formation (Fig. 4e). These results demonstrate that increased energy metabolism in MB neurons is not only necessary, but also sufficient for LTM formation. The expression of PDK RNAi had no effect on memory formed after massed training (Supplementary Fig. 2E), highlighting that the facilitative effect is specific to LTM. It had no effect either on the LTM score after regular five spaced cycles-training (Supplementary Fig. 2E), which is consistent with the increased energy metabolism having a gating effect at an early stage of LTM process (see Discussion).

Finally, we wondered if manipulating the level of mitochondrial activity in MB neurons had an effect on the feeding behaviour. In the dye-feeding assay, naive flies expressing the RNAi against PDHE1β, which downregulates mitochondrial

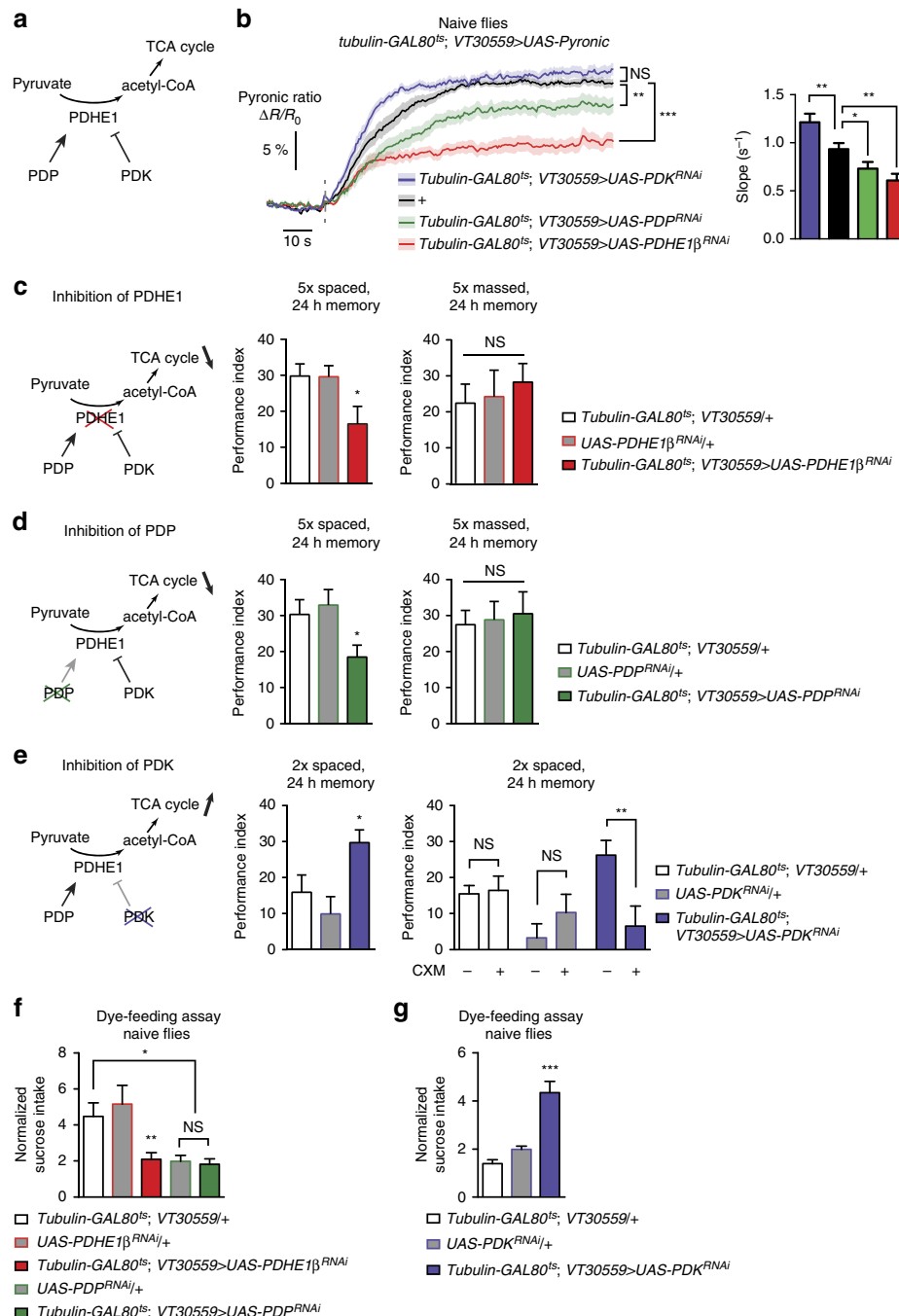

**Figure 4 | Enhanced energy flux in MB neurons following spaced training is necessary and sufficient for LTM formation. (a)** Representation of the action of PDHE1, PDP and PDK. In mitochondria, PDHE1 catalyzes the conversion of pyruvate to acetyl-coA, which enters the tricarboxylic acid cycle (TCA). PDHE1 can be inactivated through phosphorylation by PDK. In contrast, PDP can activate a phosphorylated PDHE1. **(b)** Imaging of pyruvate accumulation following azide application in naive flies co-expressing Pyronic and RNAi against either the β subunit of PDHE1 (PDHE1β), PDP or PDK in MB neurons at adult stage. The plateau value corresponding to sensor saturation was significantly different between the different conditions ($n = 11$–17; $F_{3,100} = 26.8$, $P < 0.0001$). The slope of accumulation was also significantly different: pyruvate accumulation was faster with PDK inhibition, and slower with PDP and PDHE1 inhibition, compared to control flies ($F_{3,100} = 12.24$, $P < 0.0001$). **(c)** Inhibition of PDHE1β in adult MB neurons impaired 24-h memory after $5 \times$ spaced training ($n = 15$–19; $F_{2,49} = 3.89$, $P = 0.028$), but not after $5 \times$ massed training ($n = 10$–11; $F_{2,31} = 0.26$, $P = 0.77$). **(d)** Inhibition of PDP in adult MB neurons impaired 24-h memory after $5 \times$ spaced training ($n = 14$–15; $F_{2,43} = 3.95$, $P = 0.027$), but not after $5 \times$ massed training ($n = 10$–11; $F_{2,31} = 0.088$, $P = 0.92$). **(e)** Flies expressing an RNAi against PDK in MB neurons exclusively at the adult stage showed increased memory after two spaced training cycles as compared to their genotypic controls ($n = 10$–12; $F_{2,31} = 5.16$, $P = 0.012$). The memory of these flies, but not of their genotypic controls, was sensitive to CXM treatment, which is a hallmark of LTM (*tubulin-GAL80ts; VT30559/+*: $n = 17$ and 14; $t_{29} = 0.22$; $P = 0.83$; $+/UAS$-$PDK^{RNAi}$: $n = 12$ and 13: $t_{23} = 1.09$; $P = 0.29$; *tubulin-GAL80ts; VT30559/UAS*-$PDK^{RNAi}$: $n = 18$ and 19; $t_{35} = 2.87$; $P = 0.007$). **(f,g)** Inhibition of PDHE1β in adult MB neurons of naive flies resulted in a marked decrease of sucrose intake compared to the two genotypic controls **(f)**. For PDP inhibition, one of the genotypic controls showed also low sucrose intake, which hampers the interpretation of the data. ($n = 18$ groups of 5 flies; $F_{4,89} = 6.24$, $P = 0.0002$). Inhibition of PDK induced a strong increase in sucrose intake compared to the genotypic controls (**g**; $n = 12$ groups of five flies; $F_{2,34} = 28.7$, $P < 0.0001$).

pyruvate uptake, displayed a strong decrease in sucrose intake compared to their genotypic controls (Fig. 4f). Strikingly, flies expressing the RNAi against PDK, which increases pyruvate uptake by mitochondria, showed an opposite behaviour, having a much higher sucrose intake than their controls (Fig. 4g). Remarkably, these results evidence that feeding behaviour is directly regulated by the level of energy flux in the MB. Moreover, the effect observed with PDK RNAi could explain why normal flies show an increased sucrose intake after spaced training, since this protocol in itself upregulates MB energy metabolism (Figs 1 and 2).

**Defined dopaminergic input neurons activate MB energy flux.** The LTM-gating dopaminergic neurons we previously identified[18] intervene at an early stage of LTM formation. Therefore, we investigated the putative link between dopamine signalling and the observed increase in MB energy flux. First, we asked whether LTM-gating dopamine signalling is also critical in the same time window. We used the thermosensitive mutant dynamin Shibire[ts] (Shi[ts])[40], which is dysfunctional at an elevated temperature, to block the output of genetically targeted dopaminergic neurons precisely during this time window. For this, two genetic drivers were used (Fig. 5a): the first one is a split-GAL4 line (MB438B, ref. 22) whose labelling is highly specific to two dopaminergic neurons from the paired posterior lateral 1 (PPL1) cluster, known as the V1 and MP1 neurons (respectively designated as PPL1-α′2α2 and PPL1-γ1pedc in the systematic nomenclature proposed in ref. 22). The second one is a classical GAL4 driver (NP2758) that has been used in several studies to label only MP1 neurons within dopaminergic neurons[41,42]. Flies expressing Shibire[ts] through either of these drivers displayed an LTM defect when transferred to the restrictive temperature for 3 h immediately after spaced training (Fig. 5b), but not when kept at the permissive temperature (Supplementary Fig. 3A) or after massed training (Supplementary Fig. 3B). MP1 dopaminergic neurons thus play a master role in controlling LTM formation.

We postulated that signalling from MP1 neurons could trigger the upregulation of MB energy flux after spaced training. To test this hypothesis, we aimed to artificially activate these neurons in naive flies using the heat-sensitive cation channel dTrpA1 (refs 18,43), and to measure the resulting energy flux in MB neurons. On the basis of the genetic enhancers that were used to build the MB438B split-GAL4 line, we identified a LexA driver (30E11-LexA) that allows targeting of a small set of dopaminergic neurons, including MP1 neurons, independently of the GAL4/UAS system (Supplementary Fig. 3C). We then used this driver to express dTrpA1, while expressing the Pyronic sensor in MB neurons as before. When flies were subjected to thermal treatment to activate dTrpA1-expressing neurons, the rate of pyruvate consumption measured immediately afterward in MB neurons was strongly increased as compared to flies that received a similar treatment but did not express dTrpA1 (Fig. 5c, Supplementary Fig. 3D). Without thermal treatment, no difference was observed between the two conditions (Supplementary Fig. 3E). Altogether, these data suggest that following spaced training, dopamine signalling from MP1 neurons triggers increased energy consumption in MB neurons. In this case, and given the observed facilitative effect of PDK inhibition on LTM formation (Fig. 4e), the activation of MP1 neurons should facilitate LTM formation. We trained flies with only two spaced cycles of associative conditioning, interspersed by periods of heat-activation. The 24 h-memory of flies expressing dTrpA1 through either MB438B or NP2758 driver was increased as compared to their genetic controls (Fig. 5d,e). As was the case for PDK inhibition, the memory formed in flies expressing

dTrpA1 through MB438B driver was sensitive to CXM treatment (Fig. 5d), which reveals actual LTM formation with this protocol. No increase occurred in the absence of activation periods (Supplementary Fig. 3F). These results reveal that dopamine signalling from MP1 neurons, which upregulates MB energy flux, is sufficient to engage LTM formation.

**DAMB couples dopamine with MB energy flux and LTM.** It is known that flies mutant for the D1 receptor dDA1 are defective for all forms of aversive olfactory memories, including LTM[27]. Here, we expressed RNAi against dDA1 in MB neurons at adult stage and confirmed that LTM after spaced training was impaired (Supplementary Fig. 4A, Supplementary Table 1). However, when this RNAi was expressed in MB neurons and MP1 neurons were activated in naive flies, the activation of MB energy consumption was still observed (Supplementary Fig. 4B). This indicates that the action of MP1 neurons on MB energy state is mediated by a distinct receptor that specifically controls LTM formation. Two experimental observations implicate DAMB, a type 1 dopamine receptor that is highly expressed in the MB (ref. 44) and is functionally similar to mammalian D5 (ref. 45). First, damb mutant flies were defective in LTM formation (Fig. 6a), but displayed normal olfactory acuity and shock perception (Supplementary Table 1). To confirm the physiological involvement of DAMB in LTM, we expressed RNAi against DAMB in MB neurons exclusively at the adult stage. This impaired LTM performance (Fig. 6b, Supplementary Fig. 4C), but not the 24 h-memory after massed training (Fig. 6b), nor the naive odour or shock avoidance (Supplementary Table 1). Second, we observed that when DAMB expression was knocked-down in MB neurons using an RNAi construct, MP1 activation failed to increase MB energy consumption (Fig. 6c, Supplementary Fig. 4D). Furthermore, expressing RNAi against DAMB in adult MB neurons prevented the increase in neuronal energy consumption after spaced training (Fig. 6d, Supplementary Fig. 4E) that otherwise occurred in flies not expressing this RNAi (Supplementary Fig. 4F). These results establish that DAMB mediates the promoting effect of dopamine signalling from MP1 neurons on energy flux in MB neurons. Consistent with these cellular data, we also observed that DAMB knock-down in adult MB neurons failed to elicit the increased sucrose intake that is normally observed after spaced training (Fig. 6e, Supplementary Fig. 4G). This result links the extra feeding behaviour with the upregulation of energy flux in MB neurons, but it could still be that this behaviour is due to the presence of LTM itself, rather than early upstream process leading to its formation. To discriminate between these two alternatives, using the dye-feeding assay, we tested two other genetic mutants that we identified as specifically defective for LTM: crammer[46] and debra[47]. Interestingly, while as expected the damb mutant did not show increased feeding following spaced training, the two other mutants still showed the extra feeding phenotype (Fig. 6f). This is consistent with the idea that the increased sucrose intake specifically following spaced training originates from the early upregulation of MB energy metabolism, and may not be affected by manipulations that hampers LTM at later stages.

**Discussion**
In this study we investigated whether and how energy metabolism in the Drosophila MB regulates LTM formation. Our results evidence that LTM formation involves rapid and major alterations of the energy balance, measurable both at the cellular and whole body levels. Using cell-type specific knock-down of enzymes that regulate mitochondrial energy consumption, we showed that the increased energy flux in MB neurons was

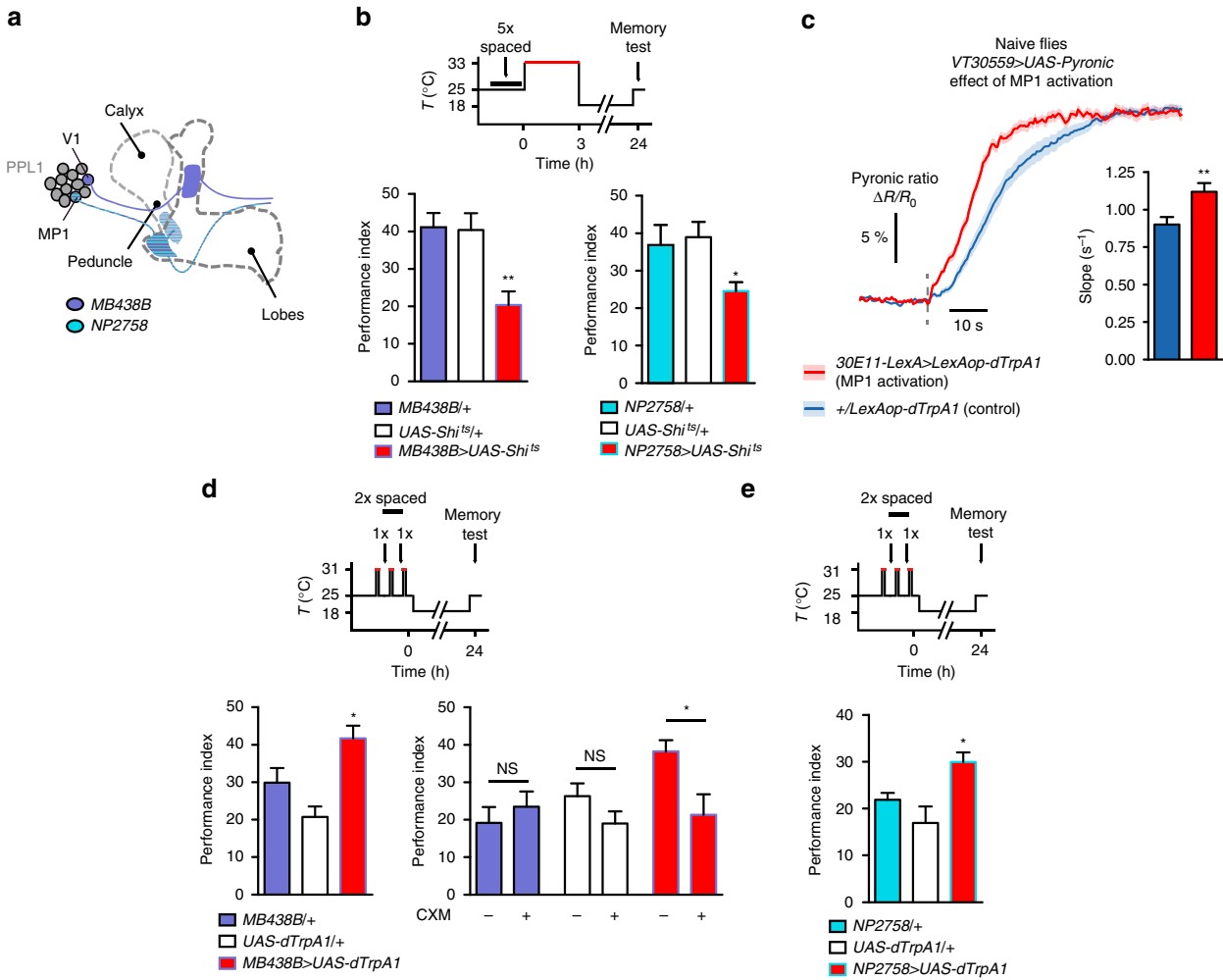

**Figure 5 | Dopamine input on MB neurons triggers increased energy flux and LTM formation. (a)** Illustration of the dopaminergic neurons in the PPL1 cluster that are targeted by the *MB438B* and *NP2758* drivers. Both drivers target MP1 neurons. Expression patterns of the driver lines used in this study are available online on several public databases. See Experimental Procedures for details. **(b)** Flies were submitted to a 3 h period at 33 °C following spaced training to block the output of Shi[ts]-expressing neurons. Flies expressing Shi[ts] through either the *MB438B* or *NP2758* driver showed an LTM defect as compared to their genotypic controls (*MB438B*: $n = 7–11$ $F_{2,26} = 9.23$, $P = 0.0011$; *NP2758*: $n = 8–11$, $F_{2,26} = 4.40$, $P = 0.023$). **(c)** Flies were subjected to a thermal treatment that consisted of two 3-min periods at 33 °C separated by 5 min. Pyruvate accumulation following azide application was measured 2–3 min later. *30E11-LexA > LexAop-dTrpA1* flies ($n = 10$) were compared to $+/LexAop-dTrpA1$ genotypic control flies ($n = 13$). The slope of pyruvate accumulation was higher in flies with activated MP1 neurons ($t_{44} = 2.87$; $P = 0.0063$). **(d)** Flies were subjected to two cycles of spaced training, interspersed by 1-min periods at 31 °C to activate dTrpA1-expressing neurons. Flies expressing dTrpA1 through the *MB438B* driver showed enhanced 24 h-memory compared to their genotypic controls (*MB438B*: $n = 8–10$, $F_{2,26} = 10.19$, $P = 0.0006$). The memory of these flies, but not of their genotypic controls, was sensitive to cycloheximide (CXM) treatment, which is a hallmark of LTM (*MB438B/+*: $n = 12$; $t_{22} = 0.73$; $P = 0.47$; $+/UAS-dTrpA1$: $n = 12$; $t_{22} = 1.56$; $P = 0.13$; *MB438B/UAS-dTrpA1*: $n = 12$; $t_{22} = 2.72$; $P = 0.012$). **(e)** Same training protocol as in **d**. Flies expressing dTrpA1 through the *NP2758* driver showed enhanced 24 h-memory compared to their genotypic controls (*NP2758*: $n = 12$, $F_{2,35} = 6.89$, $P = 0.0032$).

necessary, but also sufficient for labile memory to be consolidated into LTM. Accordingly, we established that specific dopaminergic neurons (MP1 neurons) which are required early after spaced training for LTM formation actually activate MB energy flux, through a specific dopamine receptor (the DAMB receptor). Hence, our data demonstrate an instructional role of energy metabolism pathways in LTM formation and bring forward a functional significance to dopamine signalling in the control of LTM.

We used sodium azide as an inhibitor of oxidative phosphorylation to block mitochondrial pyruvate uptake and measured the resulting intracellular pyruvate accumulation. Because glycolysis is upregulated to compensate for blocked mitochondrial activity[36], the pyruvate accumulation does not simply mirror the rate of pyruvate consumption before azide application, which

complicates the interpretation of the results. For *in vitro* applications of similar protocols, this difficulty was bypassed by depleting extracellular medium from glucose, which prevents the feedback activation of glycolysis upon azide treatment[35], but this technique cannot be used *in vivo*. However, two arguments support that our measurements indeed evaluate the mitochondrial flux. First, in naive flies, the rate of azide-evoked pyruvate accumulation in MB neurons matched the expected decreased or increased mitochondrial flux upon knock-down of regulatory enzymes in the same neurons, either activating (PDHE1, PDP) or inhibitory (PDK). Second, using a genetically-encoded glucose sensor, we observed that the glycolytic rate following azide treatment in flies after spaced training or unpaired conditioning was increased to the same level, although the rate of pyruvate accumulation was higher in the

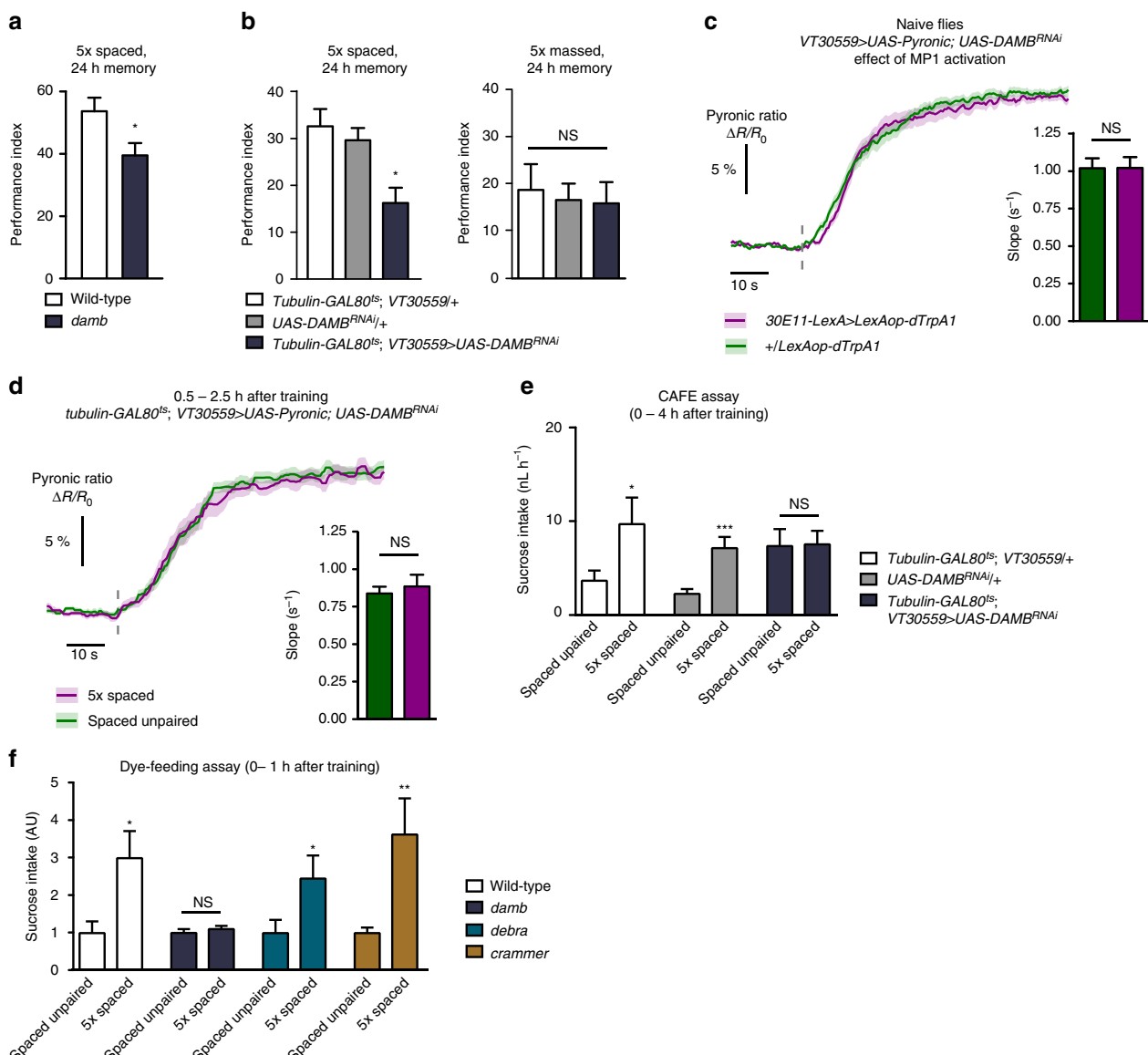

**Figure 6 | DAMB in MB neurons mediates the effect of dopamine on energy flux and LTM formation.** (**a**) *damb* mutant flies displayed an LTM defect in comparison to wild-type flies ($n = 12$–13; $t_{23} = 2.41$, $P = 0.024$). (**b**) Flies expressing RNAi against DAMB in MB neurons exclusively at the adult stage showed an LTM defect as compared to their genotypic controls ($n = 19$–21; $F_{2,60} = 7.39$, $P = 0.0014$), but exhibited normal 24 h-memory after massed training in comparison to their genotypic controls ($n = 14$; $F_{2,41} = 0.10$, $P = 0.90$). (**c**) An RNAi construct against DAMB was expressed in MB neurons. In this context, thermal treatment failed to activate MB energy flux in *30E11-LexA > LexAop-dTrpA1* flies ($n = 13$) as compared to $+/LexAop-dTrpA1$ flies ($n = 15$). No difference was measured in the slope of pyruvate accumulation between the two conditions ($t_{54} = 0.023$; $P = 0.98$). (**d**) The expression of the RNAi against DAMB at the adult stage in MB neurons suppressed the upregulated energy flux in MB neurons following spaced training ($5 \times$ spaced: $n = 10$; unpaired: $n = 10$; slope: $t_{34} = 0.54$; $P = 0.59$). (**e**) Following spaced training, flies expressing RNAi against DAMB at the adult stage in MB neurons did not present the increase in sucrose intake observed in their genotypic controls ($n = 69$–72; driver control: $t_{141} = 2.01$; $P = 0.046$; effector control: $t_{138} = 3.77$; $P = 0.0002$; flies expressing RNAi: $t_{141} = 0.085$; $P = 0.93$). (**f**) Several mutations leading to specific LTM defects were tested in the dye-feeding assay. Following spaced training, the *damb* mutant did not show an increased sucrose intake compared to the unpaired control protocol ($n = 9$; $t_{16} = 0.77$; $P = 0.45$). However, the *debra* ($n = 16$–17; $t_{31} = 2.07$; $P = 0.047$) and *crammer* ($n = 15$–16; $t_{29} = 2.77$; $P = 0.0096$) mutants still showed increased feeding behaviour like wild-type flies ($n = 24$–25; $t_{47} = 2.63$; $P = 0.012$).

former condition. This suggests that the additional pyruvate accumulation measured after spaced training does not come from an increased glycolytic flux in MB neurons. In addition, our behavioral experiments established that LTM formation is sensitive to genetic manipulations that tend to decrease mitochondrial flux, and on the contrary is facilitated when this flux is stimulated. Thus our results altogether point to a pivotal role of neuronal oxidative phosphorylation in LTM formation. The evidence reported here are consistent with an upregulation of

oxidative energy metabolism in MB neurons being an instructive signal to trigger LTM formation. Noteworthy, PDK knockdown in MB neurons did not increase memory after the regular LTM-forming five spaced cycles training protocol. On the same line, we showed in a previous study that activating a broader subset of dopaminergic neurons, including MP1 neurons, did not increase LTM after spaced training[18]. These results, together with the fact that energy metabolism is upregulated at an early stage of memory consolidation, are also consistent with the fact that the

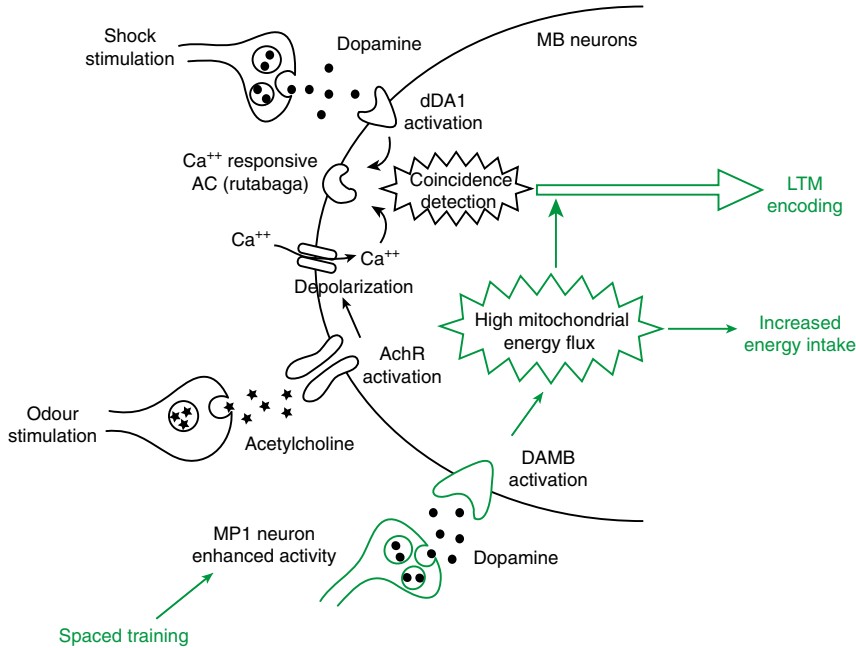

**Figure 7 | Model of energy-based gating of LTM in Drosophila.** During spaced training, the shock perception is relayed to MB neurons by dopamine neurons through the activation of the dDA1/dopR (D1) receptor (46), and olfactory input is delivered from olfactory projection neurons through cholinergic signaling. The simultaneous perception of these two stimuli activates the calcium-dependent adenylyl-cyclase (AC) rutabaga. In parallel, the enhanced activity of MP1 neurons activates the DAMB (D5) receptor, which triggers an energy flow into MB neurons. This increased energy both launches and sustains processes that lead to the formation of LTM.

metabolic shift functions as a gate to control whether LTM formation is enabled or not, but is independent from the downstream mechanisms that regulate the strength or robustness of the LTM. Hence, once the gate is open, that is, either after five spaced cycles in normal flies or after two spaced cycles in flies with stimulated energy metabolism, LTM is formed at similar magnitude, yielding similar memory scores.

The measurements of pyruvate accumulation we conducted in other structures of the fly's brain than MB did not reveal an upregulation following spaced training. Hence, the metabolic activation that we report in MB neurons is not a local manifestation of a whole brain effect, and the possibility that it constitutes an MB-specific phenomenon cannot be ruled out. Interestingly, these experiments also showed that plateau value reached by pyruvate accumulation varied between brain structures: although of similar magnitude in MB neurons and mNSC, it was much lower in EB neurons. Under the reasonable assumption that this plateau value corresponds to the saturation of the sensor, this observation betrays a higher steady-state pyruvate concentration in EB neurons. The other situation in which we observed such an effect is upon decreased PDH activity in MB neurons, which is expected to decrease mitochondrial energy metabolism and in turn activate glycolysis. Although it is difficult to draw firm conclusions with pyruvate data only, these observations suggest that distinct parts of the brain may rely on different metabolic profiles—here EB neurons being for example more glycolytic than MB and mNSC. This is an illustration of the exciting perspectives the *in vivo* use of energy metabolism sensors opens up in Drosophila.

Overall, our results support the existence of an energy switch in the MB that is triggered by dopamine signalling through the DAMB receptor to launch LTM formation (Fig. 7). In this model, dopamine acts via two receptors on MB neurons for LTM formation with distinct functional involvement: dDA1, a D1 receptor, signals the unconditioned stimulus to MB neurons for all forms of aversive memories[27], and DAMB is specifically

required to trigger LTM encoding. In mammals many studies have shown a prominent role of dopamine signalling in hippocampal-dependent memory tasks including LTM[48,49], through D1-class receptors[50–52], but without discriminating the involvement of D1 or D5 receptors. A couple of recent reports have started to investigate the role of either receptor type separately, revealing that both may actually be involved, but their respective functional roles remains to be clarified[52–54].

Importantly, the present study highlights an unsuspected role of DAMB-mediated dopamine signalling that drives MB neurons into a high-energy state. Although the downstream intracellular processes that mediate this effect remain to be unveiled, it can be expected that they have been evolutionarily conserved, so that dopamine signalling may also upregulate the fuelling ofspecific brain structures in mammals. Two well-established findings suggest that the mammalian hippocampus could likewise be functionally controlled by energy flux. First, dopaminergic neurons from the ventro-tegmental area and hippocampus form a functional loop that controls the formation of LTM[48]. Second, it was previously demonstrated that fluctuations in hippocampal extracellular glucose levels occur during the execution of demanding tasks and modulate memory performance[55,56].

In Drosophila, like in mammals, metabolic compartmentalization was reported, wherein neuronal oxidative metabolism is fuelled by glial glycolysis[57]. It would be interesting to investigate whether glial aerobic glycolysis in activated upon LTM formation in register with neuronal oxidative phosphorylation. Alternatively, because it is the major energy source for the fly brain, it may be that glial glycolysis operates close to its maximum capacity at steady state, and that LTM formation mobilizes additional energy stores such as glucose released from glial glycogenolysis[10]. Interestingly, it was proposed that the increased lactate transfer to neurons from aerobic glycolysis could serve to spare some glucose for other purpose than energy production, such as synaptic creation or remodelling[58].

Our results in Drosophila demonstrate that spaced training induces increased energy flux in MB neurons and the doubling of sucrose uptake over identical time windows, both processes involving DAMB signalling. Strikingly, we observed that bidirectional modulation of MB mitochondrial metabolism with PDHE1 or PDK knockdown, respectively, had a bidirectional effect of sucrose intake by the flies. Since DAMB signalling in MB neurons stimulates energy metabolism, it is therefore likely that the increased feeding behaviour following spaced training is a direct consequence of the upregulation of energy metabolism in MB neurons. It was shown that in Drosophila larvae, feeding behaviour is regulated by insulin signalling on MB neurons and synaptic output from MB neurons[59]. The present study suggests that this property may be extended to adult Drosophila as well.

The extra feeding being triggered by MB high-energy state, and being independent from LTM formation itself, the question of what the extra ingested food is used for and how it correlates to the amount of extra energy consumed in MB neurons remains unanswered. But the fact that LTM formation represents such a high demand provides a straightforward explanation for the groundbreaking observation that flies trained to form LTM and deprived from food and water after training exhibit premature death[60]. Previously, we also demonstrated that LTM formation is prevented when flies are starved before and after training, which is beneficial to their survival[30]. It must be stressed that the direct and strong impact of LTM formation on the need for energy intake was probably a powerful drive for the selection of such an adaptive plasticity mechanism. From a mechanistic point of view, it is noteworthy that adaptive plasticity under starvation occurs through the downregulation of MP1 neuron activity[30]. Indeed, the same neurons were recently shown to signal the ingestion of a caloric content by starved flies through the DAMB receptor in MB neurons, thereby enabling the consolidation of a labile reward memory trace into LTM[42]. These findings provide valuable food for thought regarding how the selective pressure on energy efficiency has shaped the evolution of taste and nutrient-signalling circuits on the one hand and memory networks on the other hand to result in such an integrated and protective mechanism that controls both aversive and appetitive LTM in Drosophila.

Our results show that LTM formation is sensitive to an impairment of mitochondrial flux, but notably that ARM formed after massed training is not, suggesting that specific forms of aversive memory rely on distinct brain metabolic states. In flies and in honeybees, it was shown that another behavioral trait, namely aggression, is controlled by the metabolic state of the brain, increased aggressive behaviour being causally linked to a decrease of neuronal oxidative phosphorylation[61] and a shift towards aerobic glycolysis[62]. These works and the present study altogether outline a new conceptual frame implying a tight coupling between behaviour and brain metabolic profile. There is much to find out about the functional outcomes of local metabolic plasticity[63], but in light of these pioneer studies insect research could be illuminating in the near future on these topics.

## Methods

**Fly strains.** Flies (*Drosophila Melanogaster*) were raised on standard medium containing yeast, cornmeal and agar at 18 °C and 60% humidity under a 12 h:12 h light-dark cycle (with the exception of imaging experiments: see the corresponding paragraph below). The *UAS-Shi^ts* (inserted on chromosome III) and *UAS-dTrpA1* (chr. II) lines have previously been used in multiple studies from our laboratory[18,30,64]. The *LexAop-dTrpA1* line was used in a previous study[65]. The *UAS-Pyronic* line (chr. III) and the *UAS-FLII^12Pglu-700μδ6* line were generated for the purpose of this study (see below). The *damb, crammer* and *debra* mutant lines were already published[46,47,66], the *MB438B* split-GAL4 line was used in a previous study[22], and the *30E11-LexA* line was obtained from Y. Aso (Janelia Research

Campus). The *UAS-DAMB^RNAi*, *UAS-dDA1^RNAi*, *UAS-PDHE1β^RNAi*, *UAS-PDP^RNAi* and *UAS-PDK^RNAi* lines correspond respectively to KK110947, KK102341, KK104022, GD31661 and KK106641 from the Vienna Drosophila Resource Center (VDRC); the *VT30559* GAL4 line also came from the VDRC. The *Feb170* line was obtained from Francois Bolduc (University of Alberta). All driver lines were verified by immunostainings of mCD8::GFP expression. Expression patterns are available online (http://brainbase.imp.ac.at for the VT lines, http://flweb.janelia.org/cgi-bin/flew.cgi for the JFRC lines).

To restrict UAS/GAL4-mediated expression exclusively to the adult stage, we used the TARGET system[67]: GAL4 activity was inhibited at the 18 °C raising temperature by a thermosensitive version of GAL80 expressed ubiquitously under the control of the tubulin promoter *tubulin-GAL80^ts* (inserted on chr. II), as previously reported[42]; GAL4 activity was released by transferring adult flies at 30 °C for 2–3 days.

**Olfactory conditioning and memory test.** The behaviour experiments, including the sample sizes, were conducted similarly to other studies from our research group[17,18,31]. Experimental flies were transferred to fresh bottles containing standard medium on the day before conditioning. Groups of 40–50 flies were subjected to one of the following olfactory conditioning protocols: five consecutive associative training cycles (5 × massed training), five associative cycles spaced by 15-min inter-trial intervals (5 × spaced conditioning), or two cycles spaced by a 15-min interval (2 × spaced training). Non-associative control protocols (unpaired protocols) were also employed for imaging and feeding assays. Conditioning was performed using previously described barrel-type machines that allow parallel training of up to 6 groups[31]. Throughout the conditioning protocol, each barrel was attached to a constant air flow at 2 l min⁻¹. For a single cycle of associative training, flies were first exposed to an odorant (the CS + ) for 1 min while 12 pulses of 5 s-long 60 V electric shocks were delivered; flies were then exposed 45 s later to a second odorant without shocks (the CS–) for 1 min. Odour and shocks were delivered separately during unpaired cycles, with shocks occurring 3 min before the first odorant. For the massed unpaired protocol, unpaired cycles were separated by 2-min intervals to avoid the association of shocks with the second odorant of the previous cycle. The odorants 3-octanol and 4-methylcyclohexanol, diluted in paraffin oil at 0.360 and 0.325 mM respectively, were alternately used as conditioned stimuli.

For experiments involving neuronal blockade with Shi^ts, flies were transferred to preheated bottles in a 33 °C room, immediately after the end of the last cycle of the training protocol. For experiments involving neuronal activation with dTrpA1, the barrels containing flies were attached to a 31 °C air flow at 2 l min⁻¹. For experiments involving CXM feeding, flies were transferred to vials containing filter paper soaked with 35 mM CXM in mineral water and 5% sucrose for 14–16 h before training (Fig. 4e, flies at 30 °C) or for 16–18 h before training (Fig. 5d, flies at 18°). After training and until memory test, flies were kept on regular food.

The memory test was performed in a T-maze apparatus. When tested 24 h after training, flies were kept at 18 °C between training and testing. This procedure is standard in our laboratory to obtain robust and reproducible memory scores on a 24-h range. A single performance index value is the average of two scores obtained from two groups of genotypically identical flies conditioned in two reciprocal experiments, using either odorant as CS + . The indicated '*n*' is the number of independent performance index values for each genotype.

**Feeding assays.** Measurements of food intake (CAFE assay and dye-feeding assay) were conducted on flies of a single sex to minimize variability. We chose to work on female flies to match imaging experiments (see below). Female flies were selected without anaesthesia 24 h before the experiment and kept on plain food at 18 °C and 60% humidity, except when the experiment involved thermal induction of RNAi, in which case female flies were collected 2 days before the experiment and then kept at 30 °C on plain food.

Our CAFÉ assay was adapted from ref. 68. The experiment was performed in 2 ml microtubes. Three holes were drilled in the lid of the tube: two 0.5 mm-diameter holes for air exchange and one 2.0 mm-diameter hole in which a truncated 200-μl pipette tip was inserted. The pipette tip was used to maintain a calibrated glass micropipette (5 μl, catalogue no. 53432-706; VWR). The glass micropipette was filled with a 5% sucrose solution in mineral water (Evian). A vegetal oil layer of ∼ 0.3 μl was superposed to minimize evaporation. In addition, a piece of cotton soaked with 0.4 ml mineral water was placed in the bottom of each tube, in order to allow *ad libitum* drinking, and to minimize evaporation. After each conditioning protocol, single flies were transferred to the prepared microtubes; capillaries pre-filled with sucrose solution were introduced once all flies were transferred. The initial length $l_0$ of sucrose solution was measured for each capillary. In each experiment, unoccupied microtubes were included as evaporation controls. Then, the microtubes were kept at 18 °C and 60% humidity. After the desired period of time, the length of solution remaining ($l_t$) in each capillary was measured. The length of sucrose solution consumed by each fly was calculated as $l_0 - l_t - l_e$, where $l_e$ is the average length of solution that had evaporated in unoccupied control tubes. A single experiment typically included eight flies per genotype and per condition. The indicated '*n*' is the number of single flies measured for each condition.

For the colorimetric experiment (dye-feeding assay), we used a previously described protocol[69]. After conditioning, flies were collected in bottles containing a cotton disk soaked with 3 ml of a 5% sucrose—100 mM sulforhodamine B solution in mineral water. Flies were then kept at 18 °C and 60% humidity. After 1 h, flies were transferred to a 2 ml microtube and frozen in liquid nitrogen. Groups of five flies were deposited in a 1.5 ml microtube and ground in 200 μl PBS. Microtubes were then centrifuged at 12,000 r.p.m. for 7 min. The supernatants were collected and centrifuged again at 12,000 r.p.m. for 7 min. Then, 100 μl of each supernatant was aliquoted to 96-well plate, and absorbance was measured at 570 nm. As a blank absorbance control, naive flies placed on a 5% sucrose solution without sulforhodamine B were treated in parallel. The mean absorbance of these controls was subtracted from the absorbance of the other samples. A single experiment typically included 3–4 groups of five flies per condition. The complete dataset from a given associative protocol was normalized to the control unpaired protocol. For measurements on naive flies (Fig. 4f,g), naive flies of the indicated genotype were placed at 30 °C for 2 days to induce RNAi expression in relevant groups, or kept at 18 °C for 2 days (non-induced flies). Sucrose intake was then measured for 1 h with the dye-feeding assay. Sucrose intake was normalized to the non-induced flies of the same genotype, to cancel inter-genotype variations that are important in this assay. The indicated 'n' is the number of groups of 5 flies measured for each condition.

**Measurements of locomotor activity.** Two to three day old female flies were selected 24 h before the assay and placed in regular food vials at 18 °C. They were then conditioned with a spaced training or a spaced unpaired protocol. In each experiment, 32 flies of each condition were assayed. Each fly was transferred to Trikinetics (www.trikinetics.com) glass tubes sealed with fly food on one side and a plug on the other side. Tubes were put in Trikinetics Drosophila Activity Monitors. These monitors were housed in a temperature-controlled incubator at 18 °C under constant light. Beam breaks were recorded at 1-min intervals. Activity scores were calculated by averaging the total number of beam breaks during a 1-h or 4-h period after conditioning.

**Immunohistochemistry experiments.** Female flies carrying the driver transgene were crossed to UAS-mCD8::GFP or LexAop-mCD8::GFP males. Before dissection, whole flies of female F1 progenies (3–4 days after eclosion at 25 °C) were fixed in 4% formaldehyde in PBT (PBS containing 1% Triton X-100) at 4 °C overnight. Brains were dissected in Drosophila Ringer solution and fixed for 1 h at room temperature (RT) in 4% formaldehyde in PBT. Samples were then rinsed three times for 20 min in PBT, blocked with 2% bovine serum albumin in PBT for 2 h and incubated with primary antibodies at 1:400 (rabbit anti-GFP (catalogue number: A11122), Invitrogen Molecular Probes) and 1:100 (mouse anti-nc82, DSHB, (catalogue reference: nc82)) in the blocking solution at 4 °C overnight. After rinsing, brains were incubated with secondary antibodies at 1:400 (anti-rabbit conjugated to Alexa Fluor 488 (A11034), anti-mouse conjugated to Alexa Fluor 594 (A11005), Invitrogen Molecular Probes) in the blocking solution for 3 h at RT. After rinsing, brains were mounted in Prolong Mounting Medium (Lifetechnology) for microscopy analysis. Images were acquired with a Nikon A1R confocal microscope. Confocal Z-stacks were acquired in 1 μm slices and imported into NIH ImageJ for analyses.

**Generation of transgenic flies.** The 2545_pcDNA3.1( − )Pyronic plasmid[35] (obtained from G. Bonvento) was digested by BamHI and BclI. The resulting 2,287 bp fragment was purified by electrophoresis and cloned into the pUAST vector digested by BglII. The correct orientation of the insert was verified by restriction.

The 2250 SIN-cPPT-PGK-FLIIP-glu700-WHV plasmid (obtained from G. Bonvento) was digested by BamHI and XbaI. The resulting 2,395 bp fragment was purified by electrophoresis and cloned into the pUAST vector digested by BglII and XbaI. The resulting construct was verified by restriction.

Transgenic fly strains were obtained by embryonic injection of the resulting vector, which was outsourced to Rainbow Transgenic Flies, Inc (CA, USA).

***In vivo* imaging of energy metabolism.** All imaging experiments were performed on female flies expressing Pyronic or the glucose sensor in MB neurons through the VT30559 GAL4 driver or in EB neurons and mNSC using the Feb170 GAL4 driver. Female flies were preferred because their bigger size makes surgery easier, as in all previous imaging work from our lab. Other transgenes that were used in addition to VT30559 and UAS-Pyronic are indicated in the figure panels. Crosses for imaging experiments were raised at 25 °C to increase the expression level of genetically-encoded sensors through the UAS/GAL4 system. However, flies were raised at 18 °C for the experiments involving GAL80ts (Fig. 4d,e), in order to avoid any leak of GAL4 activity. The two-day induction period at 30 °C drove sufficient expression of Pyronic to perform imaging. For experiments on conditioned flies, data were collected indiscriminately from 30 min to 3 h after training.

Flies were prepared for *in vivo* imaging, as described previously[30]. The proboscis was not glued, and no agarose was applied on the brain. At the end of surgery, a 90 μl-droplet of physiological solution (130 mM NaCl, 5 mM KCl, 2 mM MgCl$_2$, 2 mM CaCl$_2$, 36 mM sucrose, 5 mM HEPES-hemisodium salt, pH 7.3–7.4)

was applied on the preparation. Two-photon imaging was performed on a Leica TCS-SP5 upright microscope, equipped with a × 25, 0.95 NA water-immersion objective. Two-photon excitation of mTFP was achieved using a Mai Tai DeepSee laser tuned to 825 nm. 512 × 250 images were acquired at a rate of two images per second. In general, the image encompassed the vertical lobes of both brain hemispheres, although only one hemisphere was visible in some preparations. The emission channels for mTFP and Venus were the same, as described in ref. 29. One minute after the beginning of image acquisition, 10 μl of a 50 mM sodium azide solution (prepared in the same physiological solution) were injected into the 90 μl droplet bathing the fly's brain. Being a weak base, 50 mM azide had no detectable effect on the pH of the solution (pH = 7.42 ± 0.03 measured on three different solutions).

Image analysis was performed using a custom-written Matlab script. Regions of interest (ROI) were delimited by hand around each visible vertical lobe, and the average intensity of both mTFP and Venus channels over each ROI were calculated over time after background subtraction. The Pyronic sensor was designed so that FRET from mTFP to Venus decreases when pyruvate concentration increases[35]. To obtain a signal that positively correlates with pyruvate concentration, the inverse FRET ratio was computed, that is, mTFP intensity divided by Venus intensity. This ratio was normalized by a baseline value calculated over the 30 s preceding azide injection to give a normalized 'Pyronic ratio'. The increase in the Pyronic ratio typically had a bilevel waveform, with a linear increase from ∼10 to ∼70% of the plateau. The risetime (from $t = 0$ until reaching 70% of the plateau) and the slope (between 10 and 70% of the plateau) were determined automatically using the *statelevels*, *risetime* and *slewrate* functions in the Matlab signal processing toolbox. The indicated 'n' is the number of animals that were assayed for each condition. Traces from all hemispheres were pooled. All acquisition and analysis parameters were determined during preliminary experiments that are not included in the study.

For experiments with the FLII[12]Pglu-700μδ6 glucose sensor, the modifications to the protocol followed with Pyronic were the following: the images were acquired at 850 nm. The FRET ratio (YFP/CFP) was computed to obtain a signal positively correlated to glucose concentration. An application of cytochalasin B was made 20 s before azide application. For cytochalasin B application, cytochalasin B (Sigma Aldrich) was aliquoted in DMSO at a concentration of 2 mM. After a ten times dilution in physiological solution, 10 μl were injected into the 90 μl droplet bathing the brain to reach a final concentration of 20 μM. For unexplained reasons, the application of cytochalasin B produced a transient but marked drop of fluorescence in both channels. This caused artifactual twitches in the FRET ratio signals, which were masked on Fig. 2e. Glycolytic rates were measured as the slope of a linear fit to the data 12 s before azide injection for intrinsic glycolysis, and 20 s after for azide-evoked glycolysis.

**Statistical analyses.** All data are presented as mean ± s.e.m. The different experimental groups are determined by their genotypes, so no randomized allocation was performed, and the investigators were not blinded to the group allocation during experiments. Comparisons of the data series between two conditions were achieved by a two-tailed unpaired *t*-test. Results of *t*-tests are given as the value $t_x$ of the $t$ distribution with $x$ degrees of freedom obtained from the data. Comparisons between more than two distinct groups were made with a one-way ANOVA, followed by Newman–Keuls pairwise comparisons between the experimental group and its controls. The *t*-test and ANOVA are robust against slight deviations from normal distributions or the inequality of variances if the sample sizes are similar between groups[70], which was the case in our experiments. Therefore, we did not systematically test our data for normality or verify variance homogeneity before statistical tests. Instead, we adopted a uniform analysis strategy for all our data, as advised in ref. 70. ANOVA results are given as the value of the Fisher distribution $F_{(x,y)}$ obtained from the data, where $x$ is the number of degrees of freedom between groups and $y$ is the total number of degrees of freedom of the distribution. Statistical tests were performed using the GraphPad Prism 5 software. In the figures, asterisks illustrate the significance level of the *t*-test, or of the least significant pairwise comparison following an ANOVA, with the following nomenclature: *$P < 0.05$; **$P < 0.01$; ***$P < 0.001$; NS: not significant, $P > 0.05$).

**Data availability.** All relevant data are available from the authors.

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

## Acknowledgements

We thank Honorine Lucchi and Aurélie Lampin-Saint-Amaux for assistance in behavioral experiments and molecular biology. We thank Gilles Bonvento and L. Felipe Barros for sharing the Pyronic and FLII$^{12}$Pglu-700$\mu\delta$6 plasmids, and François Bolduc for providing the *Feb170* GAL4 line. Research in the T.P. laboratory was funded by the Agence Nationale pour la Recherche and the Labex MemoLife. E.D.T. was funded by a doctoral fellowship from the French Ministry of Research. L.S. was funded by a postdoctoral fellowship from the Deutsche ForschungsGemeinschaft (SCHE 1884/1-1).

## Author contributions

P.-Y.P., G.I. and T.P. designed the experiments; P.-Y.P., E.d.T., L.S., S.T. and G.I. performed experiments; V.G., K.-A.H. contributed new fly lines; P.-Y.P. wrote the original draft of the manuscript, which was commented and edited by all authors. P.-Y.P. and T.P. supervised the work. T.P. obtained funding for the project.

## Additional information

**Competing interests:** The authors declare no competing financial interests.

