## [Peer Review File · Nature Communications]

Reviewers' comments:

Reviewer #1 (Remarks to the Author):

This manuscript by Placias et al contends that an increase in 'energy flux' is a key element of long-term memory formation in *Drosophila*. There are several interesting ideas presented but I have a number of comments that I think it would be helpful for the authors to address.

The use of the word 'launch' in the title is too strong.

The authors should define exactly what they mean by 'energy flux'.

The authors are not careful enough with their use of the term LTM and seem to have forgotten to define it. I think LTM means protein synthesis dependent memory to them rather than avoidance behavior after 24h and this seems to be critical for their overall message.

Line 49: The authors discuss their prior gating model for LTM and cite 10 and 11. They say in relation to these papers that 'gating mechanism that ensures the labile memory trace that results from a single occurrence of the conditioning stimuli does not systemically consolidate into LTM'. I am not aware that either of these papers has any evidence in support of this idea. eg. an experiment where a 'gate' is removed and single trial training leads to LTM. Reference 11 for example fairly explicitly states that the enhancement in that study is one of ARM. In addition, there seems to be no clear reason why one would assume the role of repeated trials is to stabilize memory from a single trial, rather than it simply being a reflection of accumulating more knowledge.

Results line 72 (Figure 1): Flies eat more after spaced aversive training.

Could it be that spaced training leads to increased arousal, which in turn leads to more energy demand? I think the authors should do an experiment to monitor activity after spaced training in the presence of food. One could also argue that they appear to eat more because they are more active and so have more collisions with the café spouts. If the flies do not move more after spaced training it would strengthen their story.

Result line 86 (Figure 2)

Flies subjected to spaced conditioning show a higher pyruvate level in KCs when mitochondrial energy metabolism is stopped; a phenomenon the authors call "energy flux". As I said above this should be better defined.

It would also seem important to ascertain whether this is a whole brain effect produced by higher exploration in the 30 min on food until they are under the microscope. Higher exploration level and therefore higher 'energy flux' in the brain would seem like a possible explanation that needs to be ruled out.

These arguments might be addressed in some respect if the authors performed some negative and/or positive controls. A possible negative control might be analyzing 'energy flux' in another part of the brain not thought to be involved in LTM formation. It might not be the author's intention but at the current version of the manuscript I think the authors are concluding that increased energy flux is specific to the MB? What would happen to 'energy flux' in a fly that was very active?

Result in line 111 (Figure 3)

The authors manipulate the "energy flux" system.

They show that in 5x training an assumed downregulation of the energy flux leads to less LTM whereas an upregulation leads to enhancement of 2x spaced training. What happens to the 5x spaced memory when the system is manipulated to upregulate energy flux?

I found some of the other rationale to be confusing and think the terminology is not helping. In Line 77 the authors write: "...the massed training protocol (Fig. 1A, B), which is a distinct repeated associative paradigm that does not yield LTM formation...". In Figure 3b they show that massed training leads to an avoidance behavior similar to the one produced by spaced training. In their words it is not LTM because it is not protein synthesis dependent. However, in Figure 3 d they show the enhancement of 2x trial 24h memory by increasing the "energy flux". The authors need to be consistent with the definition of LTM because the experiment in 3d does not discriminate between enhancing LTM or ARM. There is also no massed trained control presented.

This seems like a recurring and critical issue with the current manuscript because looking at the title and the subheading "Increased energy flux in MB neurons is necessary and sufficient for LTM formation" it appears to be the key point of the authors wish to make. However, without showing that the induced 24h memory after 2x trials is protein synthesis dependent or showing that the 5x training with increased "energy flux" is increasing test performance the claims of LTM are not strongly supported by the data.

I am not an expert in cellular metabolism but I do wonder what altering pyruvate metabolism really does to the cell. The authors do not inform the reader of why they would consider pyruvate an important point to measure and why it would be particularly informative and suitable to manipulate. Wouldn't it be expected to have a broad impact on the cell? Isn't pyruvate required for certain amino acid synthesis? Could that be why one disrupts LTM? Why is this 'energy flux'? I would like to see some more analyses of the effects of the manipulations. It would for example seem worthwhile to test whether feeding is altered in the very same groups used in figure 3. I assume flies should eat way more if the 'energy flux' in the brain is increased and vice versa. In addition, could the authors use the sensor with the various manipulations to show that they were observing some of the expected effects?

Result line 132 (Figure 4)

The authors show that post training block of MP1 neurons inhibits LTM formation. Further activating the MP1 neuron increases the "energy flux" in KCs. Activating MP1 during 2x spaced training enhances 24 h memory. These experiments again need to be clarified with respect to whether the memory being assayed is really LTM.

The authors previously demonstrated that MP1 is involved in consolidation of appetitive memory formed by caloric sugars via the DAMB receptor (Musso et al 2015). Here they show it is involved in aversive memory consolidation and also related to food intake. I think this parallel is interesting and the authors should provide some explanation of how it might work in either scenario. Perhaps more importantly, it is not clear to me how all of these ideas relate to normal assays of LTM. If I understand the rationale correctly, flies that are starved after LTM training should not form LTM. This should be tested and discussed if the normal protocols do not involve feeding afterwards.

It is notable that the authors do not show a 5x spaced experiment with MP1 dTRPa1 stimulation. However in their prior work, Placais 2012 Figure 5d they showed that activation of TH-Gal neurons in 5x spaces did not increase memory. Moreover, activating during massed training decreased 24h memory performance (Figure 5c Placais 2012). At face value these previous results do not appear to fit with the current data or story. According to the current frame-work, enhancing the energy flux should 'launch' a non-LTM memory into LTM this is not the case for the massed training protocol and it doesn't further enhance spaced training LTM. This needs to be resolved/discussed.

Results line 166 (Figure 5)

The authors show that the DAMB receptor is important for 24 memory and for the enhanced "energy flux" in KCs mediated by the MP1 neuron as well as for more eating after learning. There is something interesting in the interaction of MP1 with KC via DAMB but it does not have to be direct effect, rather than via a shift in activity of the MB network.

Discussion

Line 190-193

The authors need to consider the previous comments regarding separating ARM and LTM before they can make strong arguments about LTM.

Line 223

As stated above, the observations could result from spaced learning leading to a certain state in which flies increase exploration, which leads to higher brain activity, which leads to more energy demand in the brain and therefore more food consumption.

Reviewer #2 (Remarks to the Author):

Information processing is energetically expensive. Evolution has provided sophisticated tissue-specific cellular and molecular engineering to ensure energy is generated in timely fashion so that neurons do not run out of steam when their output is most needed. Failure to comply with energy demand has dire consequences, as exemplified by neuronal dysfunction and death caused by hypoxia or hypoglycemia. The biochemistry of energy metabolism is ancient, predating brains, tissues and possibly cells, so it has been only natural to assume that metabolism is just a platform for the high needs of information processing, in the way computers need a power supply. However, there is recent evidence that the relationship between metabolism and cognition may be bidirectional, more complex and more interesting than anticipated. The present work explores this exciting issue in *Drosophila melanogaster*.

Authors first show that flies double their intake of sucrose at an early stage of long term memory (LTM) formation induced by spaced training. Using a FRET sensor expressed in mushroom body neurons, an increase in the rate of pyruvate accumulation after OXPHOS blockage was recorded. Massed training and unpaired spaced training were without effect on sugar intake and pyruvate accumulation. RNAi inhibition of mitochondrial pyruvate dehydrogenase or its activator the phosphatase PDK impaired LTM formation, whereas more tellingly, stimulation of pyruvate dehydrogenase by RNAi inhibition of PDK enhanced LTM formation, showing that metabolic flux is not just necessary for also sufficient. Shock and odor avoidance responses were unaffected. Next, using a series of ingenious genetic experiments, it was established that the pathway that stimulates LTM, pyruvate MB accumulation, and sugar intake, involves a specific subset of dopaminergic neurons and a specific receptor, DAMB.

This is very original and elegant research. The manuscript is concisely written, experiments are well designed and informative and the data are convincing. This is the first time the metabolic flux of identified cells is monitored in real time in *Drosophila*. By contributing compelling evidence in support of the view that metabolism can control neuronal signaling and high level brain functions, this article should have a strong impact in the neurosciences. There are a few aspects to be perfected regarding some data interpretation and discussion of the literature. To increase the visibility and impact of their effort, authors may want to consider the following points:

1. Lines 43-46 and/or Discussion. The Suzuki study was important but also was parallel work by Newman doi:10.1371/journal.pone.0028427 and a previous study by Gibbs DOI:10.1002/glia.20377. Other relevant papers regarding controlling (signaling) roles for energy metabolism in the brain are: Barros doi:10.1016/j.tins.2013.04.002, Lauritzen doi: 10.1093/cercor/bht136., Tang doi:10.1038/ncomms4284, Bozzo doi.org/10.1371/journal.pone.0071721, and Yang doi:10.1073/pnas.1322912111.

2. Line 101. Interpretation of pyruvate accumulation. A highly diffusible OXPHOS blocker like azide will stop mitochondrial pyruvate consumption on its tracks, but will also induce an instant activation of glycolysis (e.g. Bittner doi:10.3389/fnene.2010.00026) resulting in increased glycolytic pyruvate production. Thus, the rate of pyruvate accumulation in azide is not really a measure of actual flux but a measure of the maximum capacity of glycolysis. Still, it is an informative parameter, analogous to the rate of oxygen consumption measured in the presence of a mitochondrial uncoupler, which is used routinely as a parameter of the maximum mitochondrial respiratory capacity. Thus, the reference to flux should be revised throughout the manuscript. Also, it would be nice to see an image of MB neurons expressing Pyronic, perhaps in Fig. 2.

3. Lines 210-222 Possible mechanisms downstream of dopamine signaling. This section may be strengthened by discussing a recent observation in *Drosophila* of stronger glial relative to neuronal glycolysis (Volkenhoff doi.org/10.1016/j.cmet.2015.07.006), which means that the increased glycolytic capacity revealed in the present study may be partly localized to glial cells. Along the same line, a discussion of the work of Marcus Raichle and colleagues regarding aerobic glycolysis and synaptic remodeling seems warranted (e.g. Goyal doi: 10.1016/j.cmet.2013.11.020.) The signaling roles of lactate mentioned in 1 are also relevant here.

4. Line 539. Pyronic data analysis. It would be informative to know whether there were differences in the maximum change in Pyronic ratio across conditions. Inspection of the traces in Fig. 2B appears to suggest that the maximum change for the red trace (5x spaced) may be smaller than the controls. If examination of longer time courses confirms such perception, it would mean that the steady-state concentration of pyruvate is higher, providing indirect evidence for glycolytic activation. It would also mean that the difference in slope has underestimated the actual difference in pyruvate accumulation. Taking advantage of the convincing sensor saturation achieved with azide and the uncertainty of the baseline, a more robust treatment may be to normalize the data at saturation point, as done for example in a recent paper using a related lactate sensor (see Fig. 4A in Machler doi: 10.1016/j.cmet.2015.10.010). Slopes may then be re-calculated.

5. Discussion. Recent work by Gene Robinson's laboratory also points to metabolism as a controlling factor upstream of information processing and behavior in fly and honey bee (Li-Byarlay doi: 10.1073/pnas.1412306111. Barros doi: 10.1016/j.tins.2014.11.005. Chandrasekaran DOI: 10.1111/gbb.12201).

6. Dopamine has been shown to directly target glial cell energy metabolism (Requardt doi: 10.1111/j.1471-4159.2010.06940.x.), is DAMB expressed in *Drosophila*'s glia?

L. Felipe Barros

Reviewer #3 (Remarks to the Author):

In this manuscript, Preat and colleagues follow-up on their previous previous work associating LTM formation with energy costs. In the present study, they demonstrate that spaced training (which

induces LTM) induces increased feeding; that pyruvate levels rise in mushroom body neurons following spaced training; that manipulating pyruvate metabolism in both directions has reciprocal impacts on LTM formation; that dopaminergic neurons innervating the MB act through the receptor DAMB to increase pyruvate metabolism; and that the activity of these DA neurons is both necessary and sufficient for LTM induction. Overall I found this paper to be well put together. The data is all very clear and logically presented in the text. Each experiment is well designed, and, for the most part, supports the conclusions made. My only serious conceptual concern is the case for increased energy flux being instructive, rather than simply permissive, in LTM formation. This argument rests on: i) acute manipulations of DA neuron activity, which are shown to affect pyruvate consumption, but also have other roles in LTM; and ii) more chronic manipulations of pyruvate metabolism in MB neurons. It would be much nicer to acutely manipulate pyruvate metabolism during LTM formation; however, I recognize that the authors are working under the technical constraints of the system and this may not be possible. Despite this reservation, in my view the authors have made a substantial case for the involvement of energy flux in LTM. The novelty of this finding, combined with the high quality of the work, leads me to support publication, providing that some relatively minor concerns are addressed:

1) As someone not in the memory field, I found the introduction to be lacking in certain fundamental details. Most notably, the distinctions between LTM and ARM (from massed training) were not discussed, despite this being of enormous impact to the paper. For example, the fact that LTM requires protein synthesis, but ARM does not, is worthy of noting. As is the fact that massed training results in persistent memory (ARM), but that this is distinct from LTM. The current presentation of massed training simply not inducing LTM is confusing when the reader reaches figure 3 and sees that massed training does induce a persistent memory. It would also be helpful to have a diagram in figure 1 to distinguish spaced- and massed-training, as well as the unpaired conditions. Moreover, the introduction should mention that starvation blocks aversive LTM formation. I wondered about this very obvious experiment until I read the discussion and realized I had to look back at their previous paper. Thus, in general, the beginning of the paper should be written more accessibly for people outside the fly memory field.

2) The feeding experiments are all normalized to the unpaired condition, which precludes direct comparisons between feeding following massed training vs spaced training. It would be more informative to report feeding levels in a meaningful unit (e.g. nL in the CAFE assay), or at least normalize the results from each assay to only one standard, so that direct comparisons can be made between conditions.

3) Surprisingly, the VT30559 driver is not characterized, or even described, in any way. One can gather that it drives expression in the MB, but evidence of this should be given. Since, to my knowledge, this is not a widely used Gal4 line, its expression needs to be carefully characterized. From looking at the VDRC website, it's clearly expressed in the MB, but is it only expressed in Kenyon Cells? Is it expressed in all lobes? This is essential information for interpreting the experiments.

4) At line 80, the statement is made that, since flies in the feeding assays had free access to water, their sucrose consumption reflected only nutrient drive. I find this to be an unsubstantiated assumption, since I am not aware of any evidence that water deprivation does not increase consumption of aqueous sucrose in the presence of another water source. The authors should either provide this evidence, or provide evidence that flies do not increase water consumption following spaced training. Along the same lines, it would be useful to determine whether spaced training induces a non-specific increase in consumption, or whether it specifically increases sugar drive. Testing a variety of nutrients (amino acids, salts, sugars with different nutritional values and sweetness, etc.) would more firmly establish the feeding phenotype following spaced training.

5) The model that the increased feeding is specifically due to the energy costs in the MB of LTM is supported by the block in increased feeding following DAMB RNAi in Figure 5. However, since this manipulation also blocks LTM, it's still possible that the feeding change is a result of the memory itself, rather than the process of LTM formation. To strengthen your model, it would be informative to block LTM downstream of the demonstrated energy flux, and show that increased feeding is still observed. Perhaps this could be done using Shi(ts) in the MB during training or during the feeding assay itself. Since I am not in the memory field, I don't know the exact optimal experiment; however, addressing this general point would strengthen the paper.

As a preamble we would like to thank the reviewers, both for expressing their interest in our findings and for formulating very relevant and constructive criticisms. We thoroughly revised our manuscript to address the points that have been raised, adding many additional experimental data, as detailed below. We hope that the reviewers will find this revised version adequately responds to their concerns. All text modifications appear in red in the manuscript file.

Reviewer #1 (Remarks to the Author):

This manuscript by Placais et al contends that an increase in ‘energy flux’ is a key element of long-term memory formation in *Drosophila*. There are several interesting ideas presented but I have a number of comments that I think it would be helpful for the authors to address.

The use of the word ‘launch’ in the title is too strong.

We intended to highlight the two facts that (i) increased energy metabolism is a necessary and sufficient condition to form long-term memory (LTM) and (ii) the upregulation of energy metabolism occurs early in the LTM process. “Launch” seemed to match these criteria. Following the reviewer’s request, we modified the title to remove this word.

The authors should define exactly what they mean by ‘energy flux’.

The definition is now clearly stated in the introduction. It has been removed from the title.

The authors are not careful enough with their use of the term LTM and seem to have forgotten to define it. I think LTM means protein synthesis dependent memory to them rather than avoidance behavior after 24h and this seems to be critical for their overall message.

We of course agree with the reviewer on the definition of LTM as protein-synthesis dependent memory that forms in normal flies after spaced training. To avoid confusion for non-specialist readers, our initial position was indeed not to insist about this detailed definition and the resulting distinction between anesthesia-resistant memory (ARM) and LTM, the two forms of consolidated memory in flies that can drive avoidance behavior 24h after training. This stance was obviously missed its goal since it disturbed both reviewers 1 and 3. We now clearly introduce the distinct memory phases that are relevant for the paper, in the introduction and on a diagram (Fig. 1A).

Line 49: The authors discuss their prior gating model for LTM and cite 10 and 11. They say in relation to these papers that ‘gating mechanism that ensures the labile memory trace that results from a single occurrence of the conditioning stimuli does not systemically consolidate into LTM’. I am not aware that either of these papers has any evidence in support of this idea. eg. an experiment where a ‘gate’ is removed and single trial training leads to LTM. Reference 11 for example fairly explicitly states that the enhancement in that study is one of ARM. In addition, there seems to be no clear reason why one would assume the role of repeated trials is to stabilize memory from a single trial, rather than it simply being a reflection of accumulating more knowledge.

From our point of view, the fact that massed training and spaced training, which consists in the same number of cycles, yield distinct forms of memory actually suggests that the effect of multiple training is not only a matter of accumulating more knowledge. Obviously some mechanisms are specific to the spaced learning pattern and drive LTM formation. Stated as such

in the introduction, the quoted sentence could indeed appear as an over-interpretation of our previous work, and we therefore removed it. But later in the manuscript we actually show that stimulating energy metabolism in MB neurons, or activating upstream dopamine neurons, removes a 'gate', thus facilitating LTM formation. This is displayed on the model we present on Fig. 7.

Results line 72 (Figure 1): Flies eat more after spaced aversive training. Could it be that spaced training leads to increased arousal, which in turn leads to more energy demand? I think the authors should do an experiment to monitor activity after spaced training in the presence of food. One could also argue that they appear to eat more because they are more active and so have more collisions with the café spouts. If the flies do not move more after spaced training it would strengthen their story.

Following the reviewer's suggestion, we monitored the locomotor activity of flies after spaced training and unpaired control protocol using Trikinetics device. No difference was found between the two conditions (Fig. 1E)

Result line 86 (Figure 2)

Flies subjected to spaced conditioning show a higher pyruvate level in KCs when mitochondrial energy metabolism is stopped; a phenomenon the authors call "energy flux". As I said above this should be better defined.

As stated earlier, this is now defined at the beginning of the paper.

It would also seem important to ascertain whether this is a whole brain effect produced by higher exploration in the 30 min on food until they are under the microscope. Higher exploration level and therefore higher 'energy flux' in the brain would seem like a possible explanation that needs to be ruled out.

In light of our new experiment mentioned above regarding locomotor activity, this explanation seems now unlikely to us.

These arguments might be addressed in some respect if the authors performed some negative and/or positive controls. A possible negative control might be analyzing 'energy flux' in another part of the brain not thought to be involved in LTM formation. It might not be the author's intention but at the current version of the manuscript I think the authors are concluding that increased energy flux is specific to the MB? What would happen to 'energy flux' in a fly that was very active?

We addressed this important point by performing pyruvate imaging experiments in other structures of the brain. Using the Feb170 GAL4 driver, we could image the Pyronic sensor simultaneously in the ellipsoid body (EB) and in a cluster of neurosecretory cells (NSC) from the pars intercerebralis. In both cases we found no increase in pyruvate accumulation after spaced training. Therefore, we can conclude that it is not a whole brain effect, and it cannot be ruled out that this phenomenon is specific to MB neurons. A new figure 3 is dedicated to these additional experiments.

Result in line 111 (Figure 3)

The authors manipulate the "energy flux" system. They show that in 5x training an assumed downregulation of the energy flux leads to less LTM whereas an upregulation leads to

enhancement of 2x spaced training. What happens to the 5x spaced memory when the system is manipulated to upregulate energy flux?

I found some of the other rationale to be confusing and think the terminology is not helping. In Line 77 the authors write: "...the massed training protocol (Fig. 1A, B), which is a distinct repeated associative paradigm that does not yield LTM formation...". In Figure 3b they show that massed training leads to an avoidance behavior similar to the one produced by spaced training. In their words it is not LTM because it is not protein synthesis dependent. However, in Figure 3 d they show the enhancement of 2x trial 24h memory by increasing the "energy flux". The authors need to be consistent with the definition of LTM because the experiment in 3d does not discriminate between enhancing LTM or ARM. There is also no massed trained control presented.

This seems like a recurring and critical issue with the current manuscript because looking at the title and the subheading "Increased energy flux in MB neurons is necessary and sufficient for LTM formation" it appears to be the key point of the authors wish to make. However, without showing that the induced 24h memory after 2x trials is protein synthesis dependent or showing that the 5x training with increased "energy flux" is increasing test performance the claims of LTM are not strongly supported by the data.

We agree with the reviewer that this was an important point to clarify. We reproduced our 2x-spaced-cycles experiments and tested the sensitivity of the memory that is formed following this protocol to cycloheximide feeding, which inhibits *de novo* protein synthesis. We showed that the memory of flies that express RNAi against PDK is sensitive to CXM, which confirms that these flies do form LTM (Fig. 4E). We also performed 5x massed and 5x spaced experiments and found no effect of PDK knockdown (Fig. S2E). The massed experiment confirms that this manipulation has no effect on ARM, and the 5x spaced experiment is consistent with an early gating role of energy metabolism in LTM formation, as we detail in the Discussion (lines 297–301).

I am not an expert in cellular metabolism but I do wonder what altering pyruvate metabolism really does to the cell. The authors do not inform the reader of why they would consider pyruvate an important point to measure and why it would be particularly informative and suitable to manipulate. Wouldn't it be expected to have a broad impact on the cell? Isn't pyruvate required for certain amino acid synthesis? Could that be why one disrupts LTM? Why is this 'energy flux'? I would like to see some more analyses of the effects of the manipulations. It would for example seem worthwhile to test whether feeding is altered in the very same groups used in figure 3. I assume flies should eat way more if the 'energy flux' in the brain is increased and vice versa. In addition, could the authors use the sensor with the various manipulations to show that they were observing some of the expected effects?

We performed the suggested experiments to better characterize the effect of our genetic tools. We did imaging experiments in naïve flies expressing RNAi against the three enzymes that we targeted in our behavior experiments (PDHE1 β , PDP and PDK). We measured pyruvate accumulation following sodium azide treatment, which blocks mitochondria. We observed that the rate of pyruvate accumulation was decreased with PDHE1 and PDP knockdown, and increased with PDK knockdown, which matches our expectations about the effect of these manipulations on mitochondrial energy production (Fig. 4B). We also measured sucrose intake in our dye-feeding assay. We observed that knock-down of PDHE1 β induced a decrease in sucrose intake, while PDK knock-down on the contrary increased it (Fig. 4F,G). The experiment with PDP knock-down was unfortunately not conclusive because one of the controls showed low feeding as well. The results of these feeding experiments are very interesting in

themselves because they establish a direct link between energy consumption by MB neurons and the feeding behavior of the animal. They also further connect the effect of our manipulations to energy, and not to other function such as amino acid synthesis.

Result line 132 (Figure 4)

The authors show that post training block of MP1 neurons inhibits LTM formation. Further activating the MP1 neuron increases the “energy flux” in KCs. Activating MP1 during 2x spaced training enhances 24 h memory. These experiments again need to be clarified with respect to whether the memory being assayed is really LTM.

As for PDK knock-down, we performed CXM feeding with this protocol, and we confirmed that the enhanced memory is indeed LTM (Fig. 5D).

The authors previously demonstrated that MP1 is involved in consolidation of appetitive memory formed by caloric sugars via the DAMB receptor (Musso et al 2015). Here they show it is involved in aversive memory consolidation and also related to food intake. I think this parallel is interesting and the authors should provide some explanation of how it might work in either scenario. Perhaps more importantly, it is not clear to me how all of these ideas relate to normal assays of LTM. If I understand the rationale correctly, flies that are starved after LTM training should not form LTM. This should be tested and discussed if the normal protocols do not involve feeding afterwards.

A comparison of the findings reported in the paper of Musso et al. (2015) and those reported here is made in the Discussion. We wish to highlight that in the normal assay of aversive LTM after 5x spaced training, flies are transferred to regular food vials after the conditioning protocol. This is now explicitly stated in the Methods section (line 434). Interestingly, we showed in a previous work (Plaçais et al., 2013) that flies that are starved before and after training lose the ability to form LTM. This is also mentioned in the Discussion.

It is notable that the authors do not show a 5x spaced experiment with MP1 dTRPa1 stimulation. However in their prior work, Plaçais 2012 Figure 5d they showed that activation of TH-Gal neurons in 5x spaces did not increase memory. Moreover, activating during massed training decreased 24h memory performance (Figure 5c Plaçais 2012). At face value these previous results do not appear to fit with the current data or story. According to the current frame-work, enhancing the energy flux should 'launch' a non-LTM memory into LTM this is not the case for the massed training protocol and it doesn't further enhance spaced training LTM. This needs to be resolved/discussed.

The reviewer is right in citing the experiment of TH-GAL4 activation in 5x spaced training from our previous work. Indeed we observed no increase in memory at that time, which is similar to what we observed here with PDK knock-down. However, there is no inconsistency here, as the reasoning is similar in both cases: the 5x spaced protocol in itself already “opens the gate” for LTM formation, so that other manipulations that would have the same effect, such as inhibition of the ARM pathway or stimulation of energy metabolism, do not add-up on it. More precisely, as mentioned by the reviewer, we previously showed that activating dopaminergic neurons with TH-GAL4 between the cycles of a massed training impaired ARM. Why was LTM not formed with such a protocol? It was shown in the same work that massed training results in an inhibition of dopaminergic neurons' activity. Therefore, in this experiment, the effect of thermal activation by TrpA1 and inhibition by massed training mutually canceled out so that ARM could not be formed because dopaminergic neurons were

not sufficiently inhibited, but LTM could not be formed either because dopaminergic neurons were not sufficiently activated.

Overall, we do not believe that there is a discrepancy between this work and our previous work. However, we feel the argumentation we detail here about our previous massed experiment is extremely specialized and would be very obscure to most readers except a few experts of *Drosophila* memory. Therefore we prefer not to include it in the Discussion section of the manuscript, which is already rich, to avoid confusing a broader readership.

Results line 166 (Figure 5)

The authors show that the DAMB receptor is important for 24 memory and for the enhanced “energy flux” in KCs mediated by the MP1 neuron as well as for more eating after learning. There is something interesting in the interaction of MP1 with KC via DAMB but it does not have to be direct effect, rather than via a shift in activity of the MB network.

We are not completely sure about the point the reviewer intended to make. However it seems to us that there must be a dopamine receptor mediating the effect of MP1 neurons on KCs. To challenge that the effect of DAMB is direct, we tested the involvement of another dopamine receptor, dDA1/dopR, which is also involved in LTM, among other memory phases. We showed that inhibiting dDA1 expression in MB neurons did not prevent MP1 activation to enhance MB energy metabolism, although LTM formation was impaired (Fig. S4A, B). We think that these new data further support the specific role of DAMB in regulating MB energy metabolism.

Discussion

Line 190-193

The authors need to consider the previous comments regarding separating ARM and LTM before they can make strong arguments about LTM.

As explained previously, our new data now clearly support the fact that LTM formation is facilitated.

Line 223

As stated above, the observations could result from spaced learning leading to a certain state in which flies increase exploration, which leads to higher brain activity, which leads to more energy demand in the brain and therefore more food consumption.

As detailed above, we now show that flies do not show higher locomotor activity after spaced training, and hence probably not either increased exploration.

Reviewer #2 (Remarks to the Author):

Information processing is energetically expensive. Evolution has provided sophisticated tissue-specific cellular and molecular engineering to ensure energy is generated in timely fashion so that neurons do not run out of steam when their output is most needed. Failure to comply with energy demand has dire consequences, as exemplified by neuronal dysfunction and death caused by hypoxia or hypoglycemia. The biochemistry of energy metabolism is ancient, predating brains, tissues and possibly cells, so It has been only natural to assume that

metabolism is just a platform for the high needs of information processing, in the way computers need a power supply. However, there is recent evidence that the relationship between metabolism and cognition may be bidirectional, more complex and more interesting than anticipated. The present work explores this exciting issue in *Drosophila melanogaster*.

Authors first show that flies double their intake of sucrose at an early stage of long term memory (LTM) formation induced by spaced training. Using a FRET sensor expressed in mushroom body neurons, an increase in the rate of pyruvate accumulation after OXPHOS blockage was recorded. Massed training and unpaired spaced training were without effect on sugar intake and pyruvate accumulation. RNAi inhibition of mitochondrial pyruvate dehydrogenase or its activator the phosphatase PDP impaired LTM formation, whereas more tellingly, stimulation of pyruvate dehydrogenase by RNAi inhibition of PDK enhanced LTM formation, showing that metabolic flux is not just necessary for also sufficient. Shock and odor avoidance responses were unaffected. Next, using a series of ingenious genetic experiments, it was established that the pathway that stimulates LTM, pyruvate MB accumulation, and sugar intake, involves a specific subset of dopaminergic neurons and a specific receptor, DAMB.

This is very original and elegant research. The manuscript is concisely written, experiments are well designed and informative and the data are convincing. This is the first time the metabolic flux of identified cells is monitored in real time in *Drosophila*. By contributing compelling evidence in support of the view that metabolism can control neuronal signaling and high level brain functions, this article should have a strong impact in the neurosciences. There are a few aspects to be perfected regarding some data interpretation and discussion of the literature. To increase the visibility and impact of their effort, authors may want to consider the following points:

1. Lines 43-46 and/or Discussion. The Suzuki study was important but also was parallel work by Newman doi:10.1371/journal.pone.0028427 and a previous study by Gibbs DOI:10.1002/glia.20377. Other relevant papers regarding controlling (signaling) roles for energy metabolism in the brain are: Barros doi:10.1016/j.tins.2013.04.002, Lauritzen doi:10.1093/cercor/bht136., Tang doi:10.1038/ncomms4284, Bozzo doi.org/10.1371/journal.pone.0071721, and Yang doi:10.1073/pnas.1322912111.

We apologize for omitting these important contributions. We modified the first paragraph of the introduction and cited these works (lines 43–47).

2. Line 101. Interpretation of pyruvate accumulation. A highly diffusible OXPHOS blocker like azide will stop mitochondrial pyruvate consumption on its tracks, but will also induce an instant activation of glycolysis (e.g. Bittner doi:10.3389/fnene.2010.00026) resulting in increased glycolytic pyruvate production. Thus, the rate of pyruvate accumulation in azide is not really a measure of actual flux but a measure of the maximum capacity of glycolysis. Still, it is an informative parameter, analogous to the rate of oxygen consumption measured in the presence of a mitochondrial uncoupler, which is used routinely as a parameter of the maximum mitochondrial respiratory capacity. Thus, the reference to flux should be revised throughout the manuscript. Also, it would be nice to see an image of MB neurons expressing Pyronic, perhaps in Fig. 2.

As requested, we added images of MB neurons on Fig. 2, and of EB and mNSC on Fig. 3. The possibility of an activation of glycolysis induced by azide treatment indeed complicates the interpretation of pyruvate accumulation. To clarify this point, we performed important

additional experiments. Using a genetically-encoded glucose sensor that we expressed in MB neurons, we adapted a protocol describe in Bittner et al., 2010, and observed that azide indeed activates glycolysis. However, the azide-induced glycolytic rate was not increased after spaced training (Fig. 2E, F), which suggests that the faster pyruvate accumulation stems from another non-glycolytic source of pyruvate, maybe lactate, that before azide treatment fuels mitochondria. In addition, we also present new data showing the effect of RNAi against PDHE1, PDP and PDK on pyruvate accumulation following azide treatment in naïve flies (Fig. 4B). We observed that the slope of pyruvate accumulation compared to control flies matched the expected decrease (for PDHE1 and PDP knockdown) or increase (for PDK knockdown) in mitochondrial flux.

Altogether, these new data bring additional clue that our measurements reflect an upregulation of mitochondrial energy flux in MB neurons following spaced training, which is consistent with the behavioral data of Figure 4. For language accuracy, though, we limited our use of “energy flux” and replaced most occurrences with “pyruvate accumulation”. In addition we added a paragraph in the Discussion section to expose this interpretation issue and why we favor the hypothesis that oxidative metabolism is increased in MB neurons after spaced training.

3. Lines 210-222 Possible mechanisms downstream of dopamine signaling. This section may be strengthened by discussing a recent observation in *Drosophila* of stronger glial relative to neuronal glycolysis (Volkenhoff doi.org/10.1016/j.cmet.2015.07.006), which means that the increased glycolytic capacity revealed in the present study may be partly localized to glial cells. Along the same line, a discussion of the work of Marcus Raichle and colleagues regarding aerobic glycolysis and synaptic remodeling seems warranted (e.g. Goyal [doi: 10.1016/j.cmet.2013.11.020](https://doi.org/10.1016/j.cmet.2013.11.020).) The signaling roles of lactate mentioned in 1 are also relevant here.

It is indeed possible that some upregulation of glycolysis in glial cells that could co-occur to the upregulation we observe in MB neurons. Although this subject goes beyond the scope of the paper, we added a paragraph in the Discussion section about this possibility. The works of Volkenhoff et al. and Goyal et al. are cited in this paragraph.

4. Line 539. Pyronic data analysis. It would be informative to know whether there were differences in the maximum change in Pyronic ratio across conditions. Inspection of the traces in Fig. 2B appears to suggest that the maximum change for the red trace (5x spaced) may be smaller than the controls. If examination of longer time courses confirms such perception, it would mean that the steady-state concentration of pyruvate is higher, providing indirect evidence for glycolytic activation. It would also mean that the difference in slope has underestimated the actual difference in pyruvate accumulation. Taking advantage of the convincing sensor saturation achieved with azide and the uncertainty of the baseline, a more robust treatment may be to normalize the data at saturation point, as done for example in a recent paper using a related lactate sensor (see Fig. 4A in Machler [doi: 10.1016/j.cmet.2015.10.010](https://doi.org/10.1016/j.cmet.2015.10.010)). Slopes may then be re-calculated.

We re-examined the former Figure 2B: the trace with the smaller maximum change was actually the unpaired control (blue trace). The red trace (5x spaced) lies in between unpaired and 5x massed flies. But overall, there were no significant differences in the plateau values between the three conditions. We therefore kept our initial presentation for these data.

This said, we did observe significant differences in the plateau values in the experiments with RNAi against the PDH and regulatory enzymes (Fig. 4B). On that case, we followed the reviewer’s advice and presented data normalized to the maximum value to highlight the

difference in steady-state concentration. However, we feel that the gradual increase in the slope of accumulation is best displayed by the regular normalization, so we kept this for the main figure (Fig. 4B) and present the plateau normalization on the corresponding supplementary figure (Fig. S2A).

5. Discussion. Recent work by Gene Robinson's laboratory also points to metabolism as a controlling factor upstream of information processing and behavior in fly and honey bee (Li-Byarlay doi: 10.1073/pnas.1412306111. Barros doi: 10.1016/j.tins.2014.11.005. Chandrasekaran DOI: 10.1111/gbb.12201).

We thank the reviewer for bringing these very interesting studies to our knowledge. We now finish the Discussion with a paragraph about metabolic plasticity in insects, where these papers are cited.

6. Dopamine has been shown to directly target glial cell energy metabolism (Requardt doi: 10.1111/j.1471-4159.2010.06940.x.), is DAMB expressed in *Drosophila*'s glia?

To our knowledge, there are no published data about DAMB expression in glial cells. In this work we focused on neurons, and in particular MB neurons. But the question of dopamine activation of glial cells is of course of utmost interest, and it represents an attractive track for future investigation.

L. Felipe Barros

Reviewer #3 (Remarks to the Author):

In this manuscript, Preat and colleagues follow-up on their previous work associating LTM formation with energy costs. In the present study, they demonstrate that spaced training (which induces LTM) induces increased feeding; that pyruvate levels rise in mushroom body neurons following spaced training; that manipulating pyruvate metabolism in both directions has reciprocal impacts on LTM formation; that dopaminergic neurons innervating the MB act through the receptor DAMB to increase pyruvate metabolism; and that the activity of these DA neurons is both necessary and sufficient for LTM induction. Overall I found this paper to be well put together. The data is all very clear and logically presented in the text. Each experiment is well designed, and, for the most part, supports the conclusions made. My only serious conceptual concern is the case for increased energy flux being instructive, rather than simply permissive, in LTM formation. This argument rests on: i) acute manipulations of DA neuron activity, which are shown to affect pyruvate consumption, but also have other roles in LTM; and ii) more chronic manipulations of pyruvate metabolism in MB neurons. It would be much nicer to acutely manipulate pyruvate metabolism during LTM formation; however, I recognize that the authors are working under the technical constraints of the system and this may not be possible. Despite this reservation, in my view the authors have made a substantial case for the involvement of energy flux in LTM. The novelty of this finding, combined with the high quality of the work, leads me to support publication, providing that some relatively minor concerns are addressed:

1) As someone not in the memory field, I found the introduction to be lacking in certain fundamental details. Most notably, the distinctions between LTM and ARM (from massed

training) were not discussed, despite this being of enormous impact to the paper. For example, the fact that LTM requires protein synthesis, but ARM does not, is worthy of noting. As is the fact that massed training results in persistent memory (ARM), but that this is distinct from LTM. The current presentation of massed training simply not inducing LTM is confusing when the reader reaches figure 3 and sees that massed training does induce a persistent memory. It would also be helpful to have a diagram in figure 1 to distinguish spaced- and massed-training, as well as the unpaired conditions. Moreover, the introduction should mention that starvation blocks aversive LTM formation. I wondered about this very obvious experiment until I read the discussion and realized I had to look back at their previous paper. Thus, in general, the beginning of the paper should be written more accessibly for people outside the fly memory field.

As also mentioned in response to reviewer 1, we erroneously thought that not dwelling at length on the ARM vs LTM distinction would make the text more accessible to people who are not experts in *Drosophila* memory. Following this criticism, we added a diagram on Fig. 1A that details the various protocols that are used in the study, and we also expand in the introduction on the distinct forms of consolidated memory (lines 48–56). We now also cite our work on starvation and LTM as early as in the introduction (line 68).

2) The feeding experiments are all normalized to the unpaired condition, which precludes direct comparisons between feeding following massed training vs spaced training. It would be more informative to report feeding levels in a meaningful unit (e.g. nL in the CAFÉ assay), or at least normalize the results from each assay to only one standard, so that direct comparisons can be made between conditions.

We understand the reviewer's point. To allow comparisons between massed and spaced training, we converted all data obtained with the CAFÉ assay in nL per hour. However, the values of absorbance in the dye-feeding assay were more variable across genotypes and repeats of a given experiment. Although it does not change the conclusions, we thus prefer to keep the current normalization of the data, which is more relevant for comparisons between genotypes.

3) Surprisingly, the VT30559 driver is not characterized, or even described, in any way. One can gather that it drives expression in the MB, but evidence of this should be given. Since, to my knowledge, this is not a widely used Gal4 line, its expression needs to be carefully characterized. From looking at the VDRC website, it's clearly expressed in the MB, but is it only expressed in Kenyon Cells? Is it expressed in all lobes? This is essential information for interpreting the experiments.

We performed additional immunostaining experiment to further characterize the expression pattern of this driver. This driver is very specific to Kenyon cells, except for labeling in optic lobes neurons, which are not likely to play a role in olfactory memory. We also present 40x images showing that all subsets of Kenyon cells are labeled (Fig. S1A, Supplementary Movies 1 and 2).

4) At line 80, the statement is made that, since flies in the feeding assays had free access to water, their sucrose consumption reflected only nutrient drive. I find this to be an unsubstantiated assumption, since I am not aware of any evidence that water deprivation does not increase consumption of aqueous sucrose in the presence of another water source. The authors should either provide this evidence, or provide evidence that flies do not increase water consumption following spaced training. Along the same lines, it would be useful to determine

whether spaced training induces a non-specific increase in consumption, or whether it specifically increases sugar drive. Testing a variety of nutrients (amino acids, salts, sugars with different nutritional values and sweetness, etc.) would more firmly establish the feeding phenotype following spaced training.

We wish to emphasize that the aforementioned statement concerned only the CAFÉ assay. Indeed, the fact that flies are forced to stand on the capillary eat upside-down is often mentioned as a limitation of the CAFÉ assay (see for example Itskov and Ribeiro, *Front. Neurosci.*, 2013). Therefore, it seems reasonable to assume that if flies were only thirsty, they would drink water from the soaked cotton in the bottom of the tube rather than from the capillary itself. This said, we understand the reviewer's concern and we performed a control experiment with the dye-feeding assay, without sucrose in the dyes solution. No difference in dye ingestion was found between flies that underwent spaced training and their control in this case (Fig. 1D). This supports the conclusion that sucrose was indeed the motivation for increased feeding in our initial experiments.

Concerning specificity, we agree that testing other nutrients would be informative, but we think it is outside of the scope of this study, which attempts to establish a causal link between energy metabolism and LTM formation.

5) The model that the increased feeding is specifically due to the energy costs in the MB of LTM is supported by the block in increased feeding following DAMB RNAi in Figure 5. However, since this manipulation also blocks LTM, it's still possible that the feeding change is a result of the memory itself, rather than the process of LTM formation. To strengthen your model, it would be informative to block LTM downstream of the demonstrated energy flux, and show that increased feeding is still observed. Perhaps this could be done using Shi(ts) in the MB during training or during the feeding assay itself. Since I am not in the memory field, I don't know the exact optimal experiment; however, addressing this general point would strengthen the paper.

To address this very interesting point, we tested several mutant fly strains that we previously identified as specific LTM mutants in our dye-feeding assay. Strikingly, while *damb* mutant flies did not show increased feeding after spaced training, as expected from the RNAi experiment, two other mutants, *crammer* and *debra*, still displayed the increased sucrose intake phenotype (Fig. 6F). This supports our model that increased feeding results from the early stage of LTM formation, and that affecting LTM at later stages has no impact on it.

REVIEWERS' COMMENTS:

Reviewer #1 (Remarks to the Author):

Happy with revision

Reviewer #2 (Remarks to the Author):

The revision has significantly improved the manuscript. New imaging experiments examined glucose and found that memory training does not change maximum glucose consumption. This suggests that the increase in maximum pyruvate flux is sustained by lactate, perhaps from glial origin, an interesting and novel observation that resonates with the recent findings of structural metabolic differentiation between neurons and glial cells in *Drosophila*. Additional pyruvate measurements in other regions of the brain found no effect of memory training. All my other concerns have been addressed in more than adequate fashion. Congratulations for a very nice and groundbreaking study!

Reviewer #3 (Remarks to the Author):

I have reviewed the revised manuscript and the comments made in response to my (and other reviewers') concerns. In my view, the revision is substantively improved, and adequately addresses most significant issues. I have no further concerns, and support publication of this interesting and important article.